# Learning Spatial-Semantic Features for Robust Video Object Segmentation

**Xin Li**[1,2,3†], **Deshui Miao**[2†], **Zhenyu He**[2,1∗], **Yaowei Wang**[2,1∗], **Huchuan Lu**[4],
and **Ming-Hsuan Yang**[5,6]
[1]Pengcheng Laboratory [2]Harbin Institute of Technology, Shenzhen [3] Pazhou Lab (Huangpu)
[4]Dalian University of Technology [5]University of California at Merced [6]Yonsei University

## Abstract

Tracking and segmenting multiple similar objects with distinct or complex parts in long-term videos is particularly challenging due to the ambiguity in identifying target components and the confusion caused by occlusion, background clutter, and changes in appearance or environment over time. In this paper, we propose a robust video object segmentation framework that learns spatial-semantic features and discriminative object queries to address the above issues. Specifically, we construct a spatial-semantic block comprising a semantic embedding component and a spatial dependency modeling part for associating global semantic features and local spatial features, providing a comprehensive target representation. In addition, we develop a masked cross-attention module to generate object queries that focus on the most discriminative parts of target objects during query propagation, alleviating noise accumulation to ensure effective long-term query propagation. Extensive experimental results show that the proposed method achieves state-of-the-art performance on benchmark data sets, including the DAVIS2017 test (**87.8%**), YoutubeVOS 2019 (**88.1%**), MOSE val (**74.0%**), and LVOS test (**73.0%**), and demonstrate the effectiveness and generalization capacity of our model. The source code and trained models are released at https://github.com/yahooo-m/S3.

## 1 Introduction

Video Object Segmentation (VOS) aims to track and segment target objects specified in the initial frame with mask annotations in a video sequence (Xu et al., 2018; Pont-Tuset et al., 2017; Voigtlaender et al., 2019; Yang et al., 2021). This practice holds significant promise in various applications, particularly as the prevalence of video content increases in domains such as autonomous driving, augmented reality, and interactive video editing (Park et al., 2022; Bao et al., 2018). The main challenges faced by VOS include drastic target appearance change, occlusion, and identity confusion caused by similar objects and background clutter, which becomes more difficult in long-term videos.

Existing VOS methods (Oh et al., 2019; Cheng et al., 2021a) typically perform video object segmentation by comparing the test frame with past frames. Specifically, they first use an association model to generate the correlated features of the test sample and target templates. Then, the target masks are predicted based on the correlated features. The pixel-wise correlated features (Wu et al., 2023) contribute to accurate mask predictions. To account for targets that appear differently over time, some methods (Yang et al., 2021; Cheng & Schwing, 2022) utilize a memory module to store these changing appearances. In addition, several recent approaches (Wang et al., 2023; Cheng et al., 2023a) introduce object queries to help distinguish different target objects to alleviate identity confusion.

Although significant advances have been made in video object segmentation, existing methods do not perform well in challenging scenarios. First, when dealing with target objects with multiple complex or separate parts caused by occlusion, background clutter, and shape complexity, existing methods often generate incomplete prediction masks. This is because the pixel-wise correlation adopted by existing methods mainly focuses on detailed spatial information at the pixel level and neglects the semantic information at the instance level. Second, although object query improves the ID association

---

[†] Equal contribution. ∗ Corresponding author.

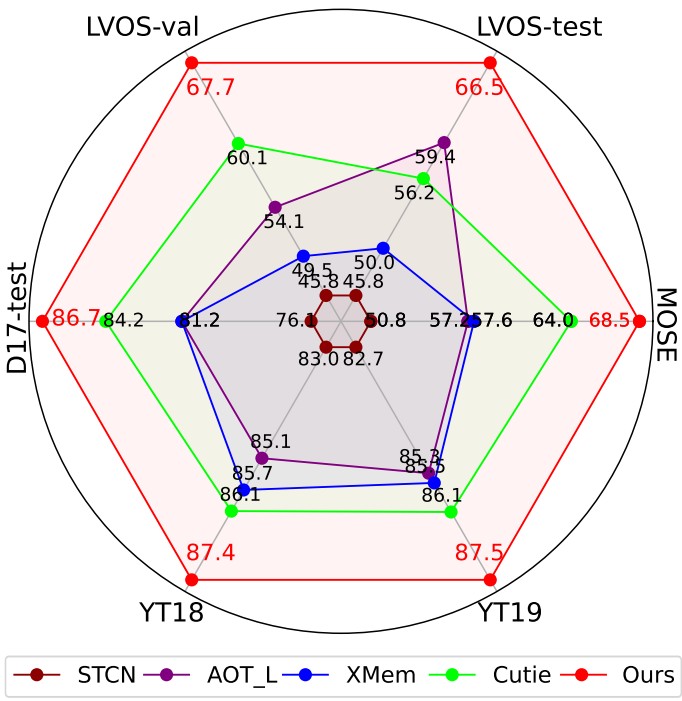

Figure 1: **Performance comparison in terms of the $\mathcal{J}\&\mathcal{F}$ score on various datasets**. The proposed method achieves favorable performance against state-of-the-art methods, including AOT_L (Yang et al., 2021), XMem (Cheng & Schwing, 2022), and Cutie (Cheng et al., 2023a) on all the datasets. All methods are trained using the YoutubeVOS and DAVIS datasets.

accuracy, it does not work well in sequences with dramatic target appearance changes. Existing methods (Wang et al., 2023; Cheng et al., 2023a) update target queries based on the entire predicted sample, which can introduce noise and errors and further lead to tracking failures when targets are missed or their identities are swapped.

To address the aforementioned issues, we propose a VOS algorithm, called S3, to learn spatial-semantic features and generate discriminative object queries for robust VOS. Specifically, we design a spatial semantic block with a semantic embedding module and a spatial dependencies modeling part. This block efficiently utilizes the semantic information and local details from pre-trained ViTs for VOS, avoiding the need to train all ViT backbone parameters. In addition, we propose a discriminative query that emphasizes representative target parts, enabling more reliable target representation and query updates. This approach mitigates noise accumulation during query propagation and enhances robustness in long-term videos. We evaluate the proposed method extensively on five benchmarks, including DAVIS 2017 (Pont-Tuset et al., 2017), Youtube VOS2018 (Xu et al., 2018), Youtube VOS2019, LVOS (Hong et al., 2023), and MOSE (Ding et al., 2023). Extensive experimental results show that our approach achieves state-of-the-art performance on all these datasets including the DAVIS2017 test (**89.1%**), YoutubeVOS 2019 (**88.5%**), MOSE (**75.1%**), and LVOS test (**73.0%**) datasets. In addition, the results achieved by only using the DAVIS and Youtube datasets for training show that our method generalizes well to different scenarios, as shown in Figure 1.

In this work, we make the following contributions:

- We propose a spatial-semantic network block to incorporate semantic information with spatial information for Video Object Segmentation. The spatial-semantic block integrates global semantic information from the CLS token of a pre-trained ViT backbone into the base features of the input sample and then models spatial dependencies using a spatial dependency modeling module, enhancing robust VOS performance when handling target objects with complex or separate parts.

- We develop a discriminative query mechanism to capture the most representative region of the target for better target representation learning and updating. This significantly improves VOS datasets, especially in long-term VOS datasets.

- We demonstrate that the proposed method achieves state-of-the-art performance on five diverse datasets and evaluate the contribution of each proposed component with comprehensive ablation studies.

## 2 RELATED WORK

**Video Object Segmentation.** Video Object Segmentation (VOS) aims to separate objects from the background and identify each target object across frames based on the target mask annotations given in the initial frame. Early VOS methods (Caelles et al., 2017; Xiao et al., 2018; Voigtlaender & Leibe, 2017; Hu et al., 2017; Luiten et al., 2018) use test-time learning to adapt pre-trained segmentation models to separate the specified targets online. The test-time learning of a deep model is time-consuming and significantly reduces the run speed. To avoid online learning, propagation-based methods (Duke et al., 2021; Cheng et al., 2018; Tsai et al., 2016) use offline shift attention to model temporal correlations between adjacent frames, propagating target masks from the previous frame to the current one. Despite promising performance, these methods suffer from error accumulation caused by inaccurate predictions. Matching-based methods (Chen et al., 2018; Hu et al., 2018; Bhat et al., 2020; Yang et al., 2020) enable fast and efficient inference by comparing the target template with the test image and predicting target masks based on feature matching. For example, OML (Chen et al., 2018) and VideoMatch (Hu et al., 2018) perform pixel matching between adjacent frames for mask prediction. Several methods (Oh et al., 2019; Yang et al., 2021; Yang & Yang, 2022; Cheng & Schwing, 2022) improve the matching-based VOS framework by introducing target memory to enrich target templates and developing advanced matching strategies to generate more comprehensive correlated features. For example, SimVOS (Wu et al., 2023) and JointFormer (Mao et al., 2021) use ViT blocks (Dosovitskiy et al., 2021) to model features and patch correspondence for object modeling jointly. For better target association across frames, some recent methods (Wang et al., 2023; Cheng et al., 2023a) introduce object queries to enrich target representation. Despite good performance on simple target objects, existing methods lack modeling of object-level semantic information, so they do not work well on dealing with objects with complex structures and long-term deformation. Unlike existing methods, the proposed approach performs pixel- and object-level feature learning through a novel object-aware backbone and a query-level correlation mechanism, enabling robust VOS performance, particularly in complex and long-term video scenarios.

**Query-Based VOS Frameworks.** DETR (Carion et al., 2020) introduces the transformer network (Dosovitskiy et al., 2021) to object detection tasks by using queries to represent target regions. In the segmentation domain, some methods (Cheng et al., 2022; Cavagnero et al., 2024) introduce masked cross-attention modules to use queries to represent targets for segmentation tasks. Motivated by the above methods, ISVOS (Wang et al., 2023) uses additional queries generated by Mask2Former (Cheng et al., 2022) to represent instance information for target modeling, enhancing target representation in the matching-based VOS framework. To enable end-to-end training, Cutie (Cheng et al., 2023a) learns object-level features and enhances the interaction between object-level and pixel-level information, leading to improved segmentation accuracy and efficiency. However, query-based VOS methods suffer from query drift due to ineffective propagation. These methods model the entire target sample in the query, incorporating redundant information that diminishes the saliency of the target object. Instead, the proposed method generates discriminative queries for target representation and propagation, which preserves key distinguishing information of target objects and offers greater robustness to variations.

**Adapters for Dense Tasks.** The ViT-Adapter (Chen et al., 2022) efficiently leverages pre-trained models by introducing a dense adapter for enhanced information exchange within ViT blocks, improving detection and segmentation performance. To enhance spatial hierarchical features, ViT-CoMer (Xia et al., 2024) introduces a multi-receptive field feature pyramid module to provide multi-scale spatial features. However, both methods require a complete fine-tuning step, which is inefficient, particularly in the video domain. Meanwhile, they focus on spatial information enhancement while lacking semantic feature interaction between ViT blocks and multi-scale features.

To better adapt to VOS tasks, the proposed method improves adapter-based VOS frameworks in the following aspects. First, we introduce a spatial-semantic module to integrate the global semantic features from ViT to multi-scale features. This module does not need extra modules to transmit information to ViT blocks, which is more efficient. Second, the proposed spatial-semantic integration block associates semantic priors with dense local features. Third, the proposed method freezes the ViT blocks during training, eliminating the need for full fine-tuning.

## 3 PROPOSED ALGORITHM

Our method aims to learn comprehensive target features that contain semantic, spatial, and discriminative information for video object segmentation. This helps us cope with complex target appearance variations and ID confusion between target objects with similar appearances in long-term videos. To this end, we propose a spatial-semantic feature learning network that first embeds semantic information from a trained ViT network with multi-scale features from a trainable CNN network. Then, we model spatial dependencies upon the fused feature, enabling spatial-semantic features for VOS. In addition, to ensure that the target query retains discriminative information during long-term appearance changes, we develop a discriminative query propagation module to capture and update local representations of the target object.

### 3.1 OVERALL FRAMEWORK

Figure 2(a) shows the overall framework of the proposed method, consisting of two core modules: a spatial-semantic feature generation module and a pixel-and-query dual-level target association module. Given a test frame $\mathbf{X}_t \in \mathbb{R}^{3 \times H \times W}$, the proposed method predicts the mask of all specified objects within it based on the initial frame $\mathbf{X}_{init} \in \mathbb{R}^{3 \times H \times W}$ and its corresponding annotation mask $\mathbf{M}_{init} \in \mathbb{R}^{H \times W}$. First, the feature generation module takes the test frame $\mathbf{X}_t$ as input and generates spatial-semantic features. Then, the target association module associates the spatial-semantic features with the reference samples pixel-wise and query-wise to better represent the correlated features. Finally, a decoder predicts the mask based on these correlated features. In addition, we develop a dual-level memory module consisting of target features and target queries to better represent video targets that vary over frames. Both kinds of memory are updated every few frames using the online predicted masks, enabling adaptive target representations for effective target association.

### 3.2 SPATIAL-SEMANTIC FEATURE LEARNING

Figure 2(b) shows the spatial-semantic network, which consists of a semantic embedding module that incorporates semantic information from a pre-trained ViT model into multi-scale features and a spatial dependency modeling module that learns spatial dependencies based on semantic features.

**Semantic embedding.** As the VOS task is designed for generic objects and no class labels are given, it is challenging to learn semantic representations directly from the VOS dataset during training. Fortunately, the CLS token in a trained ViT model aggregates semantic information from the entire image and provides a rich, global representation of the image content. Thus, we incorporate the semantic features generated from the CLS token with the multi-scale features generated from CNN networks to obtain detailed semantic features at different scales. To be specific, we first concatenate the CLS token $\mathcal{F}_{cls}^i$ with a global token representation $\mathcal{F}_g$, which is generated by averagely pooling the features from a ViT block, to generate global semantic features denoted as $(\mathcal{F}_{cls}^i, \mathcal{F}_g)$. Then, we use a cross-attention operation to generate the semantic-aware features $\mathcal{F}_{sem}^i$ by taking $(\mathcal{F}_{cls}^i, \mathcal{F}_g)$ as the *Key* and *Value*, and the spatial-temporal features $(\mathcal{F}_{spsem}^{i-1})$ from the $(i-1)$-th spatial-semantic block as the *Query*, which is formulated as:

$$\mathcal{F}_{sem}^i = f_{softmax}\left(\frac{(W^q(\mathcal{F}_{cls}^i, \mathcal{F}_g^i))(W^k\mathcal{F}_{spsem}^{i-1})^\top}{\sqrt{d}}\right)(W^v\mathcal{F}_{spsem}^{i-1}), \tag{1}$$

where $f_{softmax}$ is the SoftMax operation, and $W^q$, $W^k$, and $W^v$ are the projection matrices for *Query*, *Key*, and *Value*.

**Spatial dependency modeling.** We note that learning spatial dependencies between semantic features is important for the VOS model to understand the relationships between different object parts, which

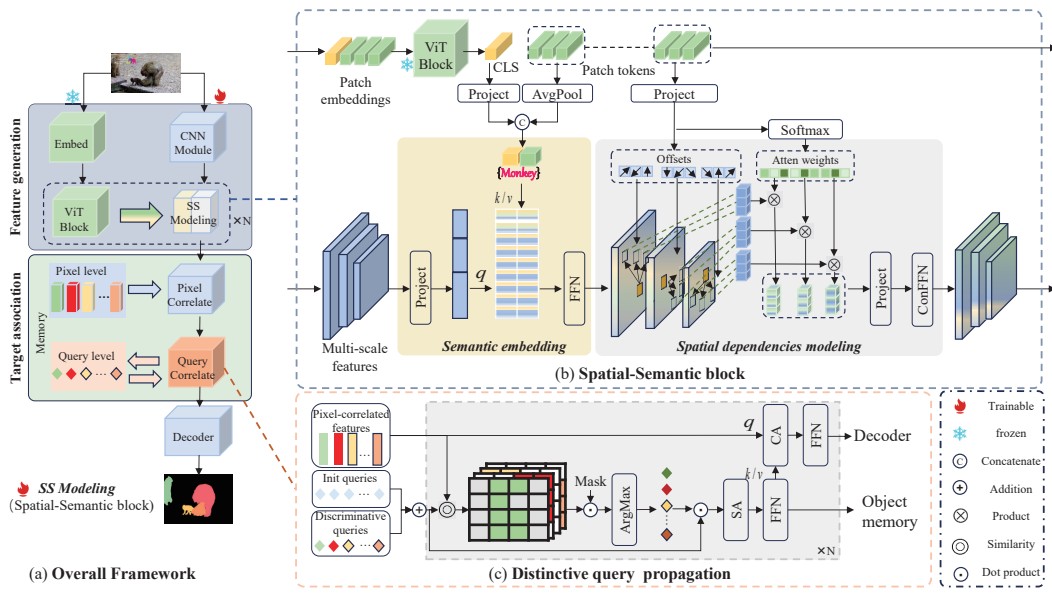

Figure 2: **Architecture of the proposed method**. (a) shows the overall framework of the proposed method comprising a feature generation part, a target association component, and a decoder-based prediction part. The feature generation part contains multiple spatial-semantic blocks, as illustrated in (b), which learn spatial-semantic features by integrating semantic priors and spatial details. (c) illustrates the discriminative query propagation process, which learns to generate discriminative queries that represent high-level object information.

helps handle objects with complex structures or separate parts. Given the features of the input image patches generated by a ViT backbone model $\mathcal{F}_{vit}^i$, we adopt the Multi-scale Deformable Cross-Attention operation (Carion et al., 2020) to capture the spatial information $\mathcal{F}_{spa}^i$ as:

$$\mathcal{F}_{spa}^i = \sum_j f_\alpha(\mathcal{F}_{vit}^i, f_{gs}(\mathcal{F}_{sem}^i, \Delta p)) \cdot f_{gs}(\mathcal{F}_{sem}^i, \Delta p), \tag{2}$$

where $\Delta p$ indicates the deformable offsets generated based on the query features $\mathcal{F}_{vit}^i$, $f_{gs}$ performs grid sampling on $\mathcal{F}_{sem}^i$ using the offsets $\Delta p$, $f_\alpha$ denotes the attention weights generated based on the query features $\mathcal{F}_{vit}^i$ and the grid sampled features, and $\sum_j$ is the summation over all sampled points. To enable an effective long-range propagation through the cascade blocks, we use the residual connection (He et al., 2016) followed by a convolution-feed-forward to generate the final spatial-semantic feature of the current block:

$$\mathcal{F}_{spsem}^i = \mathcal{F}_{spa}^i + f_{ConFFN}\left(\mathcal{F}_{spa}^i + \mathcal{F}_{spsem}^{i-1}\right). \tag{3}$$

The spatial-semantic block leverages the strengths of both a pre-trained ViT network for providing semantic information and a multi-scale CNN network for rich spatial information, enabling comprehensive spatial-semantic feature learning for robust VOS.

Figure 3(a) shows how the feature changes during the processing of the proposed model by visualizing the feature maps in different processing steps. It shows that the semantic embedding module enhances the semantic information (boundaries) of the objects in the feature maps. In contrast, after spatial dependence modeling, the feature maps pay more attention to the details of the objects. Figure 3(b) compares the features generated by different networks. Compared to ViT and ResNet models, the proposed Spatial-Semantic model exhibits better target representation capabilities with significantly higher activations in target object areas.

## 3.3 DISCRIMINATIVE QUERY PROPAGATION

To propagate target queries effectively across frames, we update the target queries with the most distinctive feature of the target object.

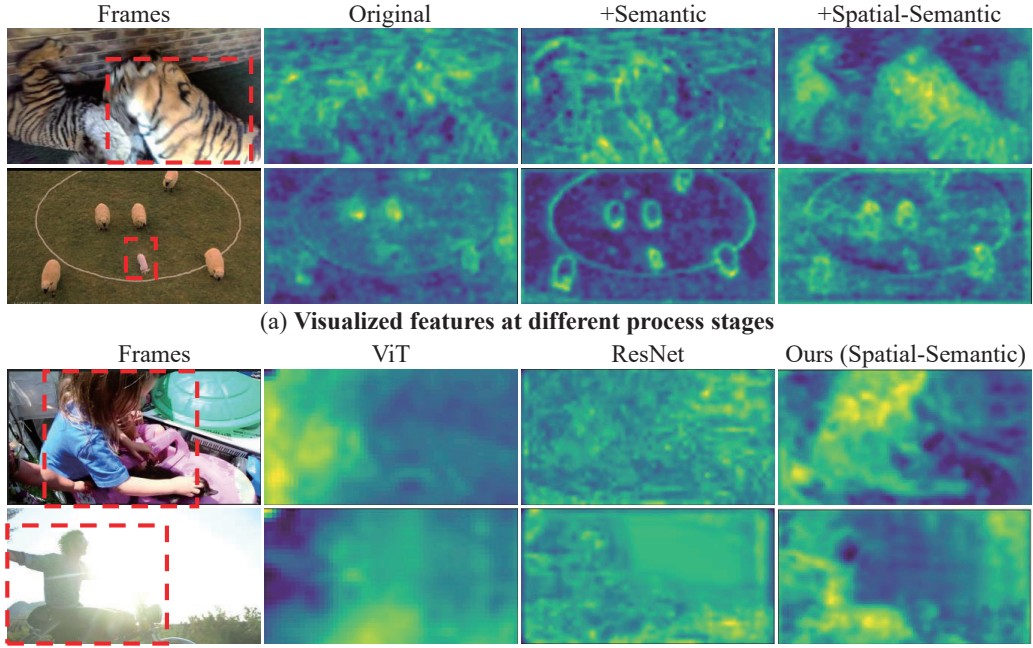

(a) **Visualized features at different process stages**

(b) **Visualization of features generated by different networks**

Figure 3: **Visualized feature maps from different backbones and stages.** It shows that the proposed spatial-semantic model generates effective features for target representation.

**Discriminative feature generation.** We select the discriminative feature of a target object by comparing the target query $\mathcal{Q} \in \mathbb{R}^{N \times C}$ with every channel activation in the correlated feature map of the target and taking the most similar one. Formally, given the feature map $\mathcal{F} \in \mathbb{R}^{H \times W \times C}$, the salient channel activation feature of a target is computed as:

$$\mathcal{Q}_s = \mathcal{F}[\arg\max_{i \in (1, H*W)} (\mathcal{F}\mathcal{Q}^T \odot \mathcal{M})], \tag{4}$$

where $\odot$ denotes the dot product, $\mathcal{M}$ represents a binary mask of the target, and $\mathcal{F}[i]$ denotes the $i$-th channel activation of the feature map $\mathcal{F}$. We use the strategy in Cutie (Cheng et al., 2023a) to generate the target mask $\mathcal{M}$, which uses the predicted segmentation results in the previous frame.

**Query propagation.** Based on the discriminative target feature $\mathcal{Q}_s$ generated from the target sample and the input target query $\mathcal{Q}_{\text{in}}$ at the current stage, the output query $\mathcal{Q}_{\text{out}}$ is propagated as:

$$\mathcal{Q}_{\text{out}} = (\alpha \odot \frac{\mathcal{A}}{\|\mathcal{A}\|_2} + \mathcal{Q}_s)\mathcal{W}_{\text{out}} + \mathcal{Q}_{\text{in}}, \tag{5}$$

where $\alpha$ is a learnable parameter to scale the matrix, $\mathcal{W}_{\text{out}}$ is a learnable projection matrix, and $\mathcal{A}$ denotes the correspondence between salient query and salient pixel features, which is computed as.

$$\mathcal{A} = \mathcal{Q} \odot \mathcal{Q}_s. \tag{6}$$

The proposed discriminative query propagation scheme adaptively refines target queries with the most representative features, which helps deal with the challenges of dramatic appearance variations in long-term videos. We also present the similarity map between the discriminative query and the feature in the last two rows, which indicates that our proposed discriminative query can represent and distinguish different objects.

## 4 EXPERIMENTS

### 4.1 EXPERIMENTAL SETTINGS

**Network.** The full version of the proposed method uses a pre-trained ViT-base (Dosovitskiy et al., 2021) model as the ViT branch and constructs a simple CNN module for multi-scale feature generation.

Table 1: **Ablation study.** All the variants are well aligned using the same settings and trained on the YoutubeVOS and DAVIS datasets. S3 denotes the proposed algorithm.

| Dataset | MOSE-val | | | DAVIS 2017 test | | | LVOS test | | | LVOS val | | | YouTube-VOS 2019 val | | | | |
| Method | $\mathcal{J}\&\mathcal{F}$ | $\mathcal{J}$ | $\mathcal{F}$ | $\mathcal{J}\&\mathcal{F}$ | $\mathcal{J}$ | $\mathcal{F}$ | $\mathcal{J}\&\mathcal{F}$ | $\mathcal{J}$ | $\mathcal{F}$ | $\mathcal{G}$ | $\mathcal{J}$ | $\mathcal{F}$ | $\mathcal{G}$ | $\mathcal{J}_s$ | $\mathcal{F}_s$ | $\mathcal{J}_u$ | $\mathcal{F}_u$ |
|---|---|---|---|---|---|---|---|---|---|---|---|---|---|---|---|---|---|
| XMem (Cheng & Schwing, 2022) (Baseline) | 53.3 | 62.0 | 57.6 | 81.0 | 77.4 | 84.5 | 50.0 | 45.5 | 54.4 | 49.5 | 45.2 | 53.7 | 85.5 | 84.3 | 88.6 | 80.3 | 88.6 |
| Cutie (Cheng et al., 2023a) | 64.0 | 60.0 | 67.9 | 84.2 | 80.6 | 87.7 | 56.2 | 51.8 | 60.5 | 60.1 | 55.9 | 64.2 | 86.1 | 85.8 | 90.5 | 80.0 | 88.0 |
| +Discriminative Query | 64.2 | 60.3 | 68.1 | 85.2 | 81.8 | 88.5 | 57.4 | 53.3 | 61.5 | 62.1 | 57.9 | 66.2 | 86.5 | 86.2 | 90.7 | 80.6 | 88.8 |
| +ViT | 64.2 | 60.2 | 68.3 | 85.6 | 82.0 | 89.2 | 58.3 | 53.7 | 62.8 | 58.9 | 54.1 | 63.6 | 86.7 | 86.4 | 90.3 | 81.0 | 88.7 |
| +Spatial-semantic (S3) | 68.5 | 64.5 | 72.6 | 86.7 | 82.7 | 90.8 | 66.5 | 62.1 | 70.8 | 67.7 | 63.1 | 72.2 | 87.5 | 86.8 | 91.8 | 81.3 | 89.9 |

The Spatial-Semantic block performs feature modeling after every block of the ViT branch. Upon the target association part, the Decoder predicts target masks based on multi-scale features. We use a soft aggregation operation for the scenario containing multiple target objects to merge the predicted masks.

In all experiments, we set the number of Spatial-Semantic Blocks to N=4.

**Training.** We adopt similar training settings as those of Cutie (Cheng et al., 2023a). Note that, apart from ResNet pre-training, we do not pre-train other models on static images and directly trained them on video datasets (YoutubeVOS (Xu et al., 2018) and DAVIS (Pont-Tuset et al., 2017)). To enhance the performance of our model, we use the MEGA dataset constructed by Cutie, which includes the YouTubeVOS (Xu et al., 2018), DAVIS (Pont-Tuset et al., 2017), OVIS (Qi et al., 2022), MOSE (Ding et al., 2023) and BURST (Athar et al., 2023) datasets. We sample 8 frames to train the model and randomly select 3 of them to train the matching process. For each sequence, we randomly choose at most 3 targets for training. Point supervision in loss is adopted to reduce the memory requirements.

For optimization, the AdamW (Kingma & Ba, 2014) optimizer is used with a learning rate of 5e-5 and a weight decay of 0.5. We train the model for 125K with a batch size of 16 on the video dataset, and 195k on the MEGA dataset. For specific parameter settings, please refer to our supplementary material. All our models are trained on a machine with 8 x NVIDIA V100 GPUs and tested using one NVIDIA V100 GPU.

**Inference.** Our feature and query memories are updated every 5th frame during the testing phase. For longer sequences, we employ a long-term fusion strategy (Cheng & Schwing, 2022) for updating. We skip frames without targets. We use two kinds of input sizes, 480 and 600/720, for more detailed comparisons. In this section, we compare all methods with input size 480p.

## 4.2 ABLATION STUDY

**Effect of every component.** Table 1 provides a comprehensive analysis of the components of our method. The first two rows describe the current methods upon which we build improvements. Cutie improves performance by using a query transformer within the model.

**Baseline**. We adopt XMem as the baseline model, a representative VOS framework with a well-designed memory mechanism.

**+ Discriminative Query**, which performs discriminative query generation by adaptively refining target queries with the most representative features of the query transformer. To validate the effectiveness of our proposed discriminative query propagation, we first choose ResNet50 as the backbone for feature generation (simply comparing query propagation with Cutie (Cheng et al., 2023a)). Performance gains compared to Cutie demonstrate the advantages of target modeling of discriminative queries for VOS, especially in long-term benchmarks.

**+ ViT**, which directly utilizes ViTDet (Li et al., 2022b) to incorporate pre-trained model features at multiple scales. The performance indicates that directly employing ViT to generate multi-scale features is minimally beneficial for VOS tasks.

**+ Spatial-semantic**, which applies the spatial-semantic blocks to integrate semantic information and multi-scale features. It is clear that the proposed spatial dependency modeling significantly boosts model performance in terms of $\mathcal{J}\&\mathcal{F}$ on all VOS benchmarks. These improvements validate the benefits of the spatial-semantic network, which improves the semantic understanding of targets.

Table 2: **Effect of different settings in every component.** The ablation study examines the impact of pre-trained weights and the number of queries on training results for the YoutubeVOS and DAVIS datasets. Additionally, it explores the effects of different adapters on the MEGA dataset.

| Dataset | MOSE-val | | | DAVIS 2017 test | | | LVOS test | | | YouTube-VOS 2018 val | | | | | YouTube-VOS 2019 val | | | | |
| --- | --- | --- | --- | --- | --- | --- | --- | --- | --- | --- | --- | --- | --- | --- | --- | --- | --- | --- | --- |
| Method | $\mathcal{J}\&\mathcal{F}$ | $\mathcal{J}$ | $\mathcal{F}$ | $\mathcal{J}\&\mathcal{F}$ | $\mathcal{J}$ | $\mathcal{F}$ | $\mathcal{J}\&\mathcal{F}$ | $\mathcal{J}$ | $\mathcal{F}$ | $\mathcal{G}$ | $\mathcal{J}_s$ | $\mathcal{F}_s$ | $\mathcal{J}_u$ | $\mathcal{F}_u$ | $\mathcal{G}$ | $\mathcal{J}_s$ | $\mathcal{F}_s$ | $\mathcal{J}_u$ | $\mathcal{F}_u$ |
| *Effect of different pre-trained weights.* | | | | | | | | | | | | | | | | | | | |
| ResNet (He et al., 2016) (Baseline) | 64.2 | 60.3 | 68.1 | 85.2 | 81.8 | 88.5 | 57.4 | 53.3 | 61.5 | 86.3 | 86.2 | 91.0 | 79.9 | 88.3 | 86.5 | 86.2 | 90.7 | 80.6 | 88.8 |
| MAE (He et al., 2022) | 67.0 | 62.6 | 71.3 | 85.3 | 81.3 | 89.3 | 64.4 | 60.2 | 68.7 | 86.4 | 85.6 | 90.2 | 80.6 | 89.0 | 86.4 | 85.4 | 89.7 | 81.2 | 89.3 |
| DINOv2 (Oquab et al., 2023) | 67.5 | 63.5 | 71.6 | 86.1 | 82.7 | 89.6 | 63.2 | 58.7 | 67.6 | 87.3 | 86.6 | 91.7 | 80.9 | 89.9 | 86.7 | 86.2 | 91.0 | 80.6 | 89.1 |
| ViT-small (Yang et al., 2024) | 64.3 | 60.0 | 68.5 | 85.3 | 81.5 | 89.1 | 55.9 | 51.6 | 60.1 | 86.0 | 85.3 | 90.4 | 79.9 | 88.5 | 86.3 | 85.3 | 90.1 | 80.6 | 89.1 |
| Depth_Anything | 68.5 | 64.5 | 72.6 | 86.7 | 82.7 | 90.8 | 66.5 | 62.1 | 70.8 | 87.4 | 87.0 | 92.0 | 80.9 | 89.7 | 87.5 | 86.8 | 91.8 | 81.3 | 89.9 |
| *Effect of different query numbers.* | | | | | | | | | | | | | | | | | | | |
| 32-query | 68.3 | 64.3 | 72.3 | 85.8 | 81.7 | 89.9 | 64.8 | 60.4 | 69.6 | 86.8 | 86.9 | 92.0 | 79.8 | 88.6 | 86.9 | 86.8 | 91.6 | 80.3 | 88.8 |
| 16-query | 67.7 | 63.6 | 71.8 | 86.6 | 82.7 | 90.2 | 66.4 | 62.0 | 70.9 | 86.9 | 86.7 | 91.8 | 80.1 | 88.9 | 87.0 | 86.6 | 91.6 | 80.6 | 89.3 |
| 8-query | 68.5 | 64.5 | 72.6 | 86.7 | 82.7 | 90.8 | 66.5 | 62.1 | 70.8 | 87.4 | 87.0 | 92.0 | 80.9 | 89.7 | 87.5 | 86.8 | 91.8 | 81.3 | 89.9 |
| *Effect of different adapters.* | | | | | | | | | | | | | | | | | | | |
| Ours (Adapter (Chen et al., 2022) ) | 72.2 | 67.8 | 76.4 | 87.2 | 84.1 | 90.4 | 71.1 | 66.7 | 75.6 | 87.1 | 86.1 | 91.0 | 81.6 | 89.8 | 87.1 | 85.9 | 90.9 | 81.4 | 90.1 |
| Ours (CoMer (Xia et al., 2024)) | 73.2 | 69.1 | 77.3 | 87.0 | 83.2 | 90.8 | 73.4 | 68.8 | 77.9 | 87.3 | 87.1 | 92.2 | 80.7 | 89.4 | 87.4 | 87.0 | 91.7 | 81.3 | 89.6 |
| Ours (Spatial-Semantic) | 74.0 | 69.8 | 78.3 | 87.8 | 84.0 | 91.7 | 73.0 | 68.3 | 77.8 | 88.0 | 87.0 | 91.8 | 82.5 | 90.7 | 88.1 | 87.4 | 92.5 | 81.9 | 90.7 |

**Effect of different pre-trained weights.** Table 2 also compares the performance of using different training parameters. We conduct experiments using three different pre-training parameters, and the results indicate that better pre-training parameters lead to more significant improvements in the model, especially those from Depth Anything (Yang et al., 2024) (8-query in Table 2). The Depth Anything model implicitly contains depth information about the scene, which contributes to its stronger representation capability, consistent with the findings of the Depth Anything paper.

**Effect of different query numbers.** Table 2 shows that performance drops as the number of queries increases. This is because more queries introduce additional background noise, given the fewer targets in most VOS benchmarks, except MOSE (Ding et al., 2023).

**Effect of adapters.** Table 2 presents a performance comparison between different adapters and our method. We replace the fusion module with various adapters and fully fine-tune them on the MEGA dataset, as indicated in related research that full fine-tuning achieves optimal performance (Chen et al., 2022; Xia et al., 2024). The experimental results demonstrate that our method is efficient in training and outperforms the other two methods in accuracy.

## 4.3 COMPARISONS WITH STATE-OF-THE-ART METHODS

**DAVIS 2017** (Pont-Tuset et al., 2017) is a benchmark that offers densely annotated, high-quality, full-resolution videos with multiple objects of interest. Table 3 shows that our method achieves favorable performance (**86.7%**) against state-of-the-art methods when trained on YoutubeVOS and DAVIS, without any pre-training. The gains are primarily due to the injection of semantic information, which helps handle sequences with object occlusion. When using the MEGA dataset as the training data, our method performs best on the DAVIS test set (**87.8%**).

**YouTubeVOS 2018** (Xu et al., 2018) contains 347 videos of 65 categories for training and videos for validation. For validation, there are 26 categories that the model has not seen in its training, allowing us to evaluate its generalization ability for 350 class-agnostic targets. Table 3 illustrates that our method performs better on seen and unseen categories than existing state-of-the-art models. The performance gain can be attributed to spatial-semantic feature representation and representative target correlation.

**YouTubeVOS 2019** is an extension of YouTube-VOS 2018, featuring a larger number of masked targets and more challenging sequences in its validation set. In Table 3, our approach exhibits competitive performance compared to the leading state-of-the-art methods. Remarkably, our model achieves the highest performance (**87.5%**) without relying on pre-training. This underscores the effectiveness of our method in leveraging inherent semantic-spatial information and discriminative query representation.

**MOSE** (Ding et al., 2023) contains 2149 video clips and 5200 objects from 36 categories under more complex scenes. Compared to existing state-of-the-art methods, our model gains great improvement in this dataset. The benefits stem primarily from the discriminative query that captures more

Table 3: **Quantitative comparisons on the MOSE, LVOS, DAVIS 2017, YouTube-VOS 2018 & 2019 dataset.** The best two results are shown in red and blue color. In the table, * denotes that the models are pre-trained using the static image datasets (Shi et al., 2016; Li et al., 2020; Zeng et al., 2019; Cheng et al., 2020; Wang et al., 2017). In the Appendix, we compare methods with external training data (e.g., large BL30K dataset for pre-training) and improved test size (600p/720p).

| Dataset | MOSE-val | | | LVOS test | | | DAVIS 2017 test | | | YouTube-VOS 2018 val | | | | | YouTube-VOS 2019 val | | | | | FPS |
|---|---|---|---|---|---|---|---|---|---|---|---|---|---|---|---|---|---|---|---|---|
| Method | $\mathcal{J}\&\mathcal{F}$ | $\mathcal{J}$ | $\mathcal{F}$ | $\mathcal{J}\&\mathcal{F}$ | $\mathcal{J}$ | $\mathcal{F}$ | $\mathcal{J}\&\mathcal{F}$ | $\mathcal{J}$ | $\mathcal{F}$ | $\mathcal{G}$ | $\mathcal{J}_s$ | $\mathcal{F}_s$ | $\mathcal{J}_u$ | $\mathcal{F}_u$ | $\mathcal{G}$ | $\mathcal{J}_s$ | $\mathcal{F}_s$ | $\mathcal{J}_u$ | $\mathcal{F}_u$ | |
| *Trained only on the YouTube VOS, and DAVIS datasets* | | | | | | | | | | | | | | | | | | | | |
| MiVOS (Cheng et al., 2021b)* | - | - | - | - | - | - | 78.6 | 74.9 | 82.2 | 82.6 | 81.1 | 85.6 | 77.7 | 86.2 | 82.4 | 80.6 | 84.7 | 78.1 | 86.4 | - |
| STCN (Cheng et al., 2021a) * | 52.5 | 48.5 | 56.6 | 45.8 | 41.1 | 50.5 | 77.8 | 74.3 | 81.3 | 84.3 | 83.2 | 87.9 | 79.0 | 87.3 | 84.2 | 82.6 | 87.0 | 79.4 | 87.7 | 13.2 |
| Swin-B-AOT-L (Yang et al., 2021) * | 59.4 | 53.6 | 65.2 | 54.4 | 49.3 | 59.4 | 81.2 | 77.3 | 85.1 | 85.1 | 85.1 | 90.1 | 78.4 | 86.9 | 85.3 | 84.6 | 89.5 | 79.3 | 87.7 | 12.1 |
| DeAOT-R50 (Yang & Yang, 2022) | 59.0 | 54.6 | 63.4 | - | - | - | 80.7 | 76.9 | 84.5 | 86.0 | 84.9 | 89.9 | 80.4 | 88.7 | 85.6 | 84.2 | 89.2 | 80.2 | 88.8 | 11.7 |
| XMem (Cheng & Schwing, 2022) | - | - | - | - | - | - | 79.8 | 76.3 | 83.4 | 84.3 | 83.9 | 88.8 | 77.7 | 86.7 | 84.2 | 83.8 | 88.3 | 78.1 | 86.7 | |
| XMem (Cheng & Schwing, 2022) * | 53.3 | 62.0 | 57.6 | 50.0 | 45.5 | 54.4 | 81.0 | 77.4 | 84.5 | 85.7 | 84.6 | 89.3 | 80.2 | 88.7 | 85.5 | 84.3 | 88.6 | 80.3 | 88.6 | 22.6 |
| ISVOS (Wang et al., 2023) * | - | - | - | - | - | - | 82.8 | 79.3 | 86.2 | 86.3 | 85.5 | 90.2 | 80.5 | 88.8 | 86.1 | 85.2 | 89.7 | 80.7 | 88.9 | 5.8 |
| SimVOS-B (Wu et al., 2023) | 61.6 | 57.9 | 65.3 | - | - | - | 82.3 | 78.7 | 85.8 | - | - | - | - | - | 84.2 | 83.1 | 87.5 | 79.1 | 87.2 | 3.3 |
| Cutie (Cheng et al., 2023a)* | 64.0 | 60.0 | 67.9 | 56.2 | 51.8 | 60.5 | 84.2 | 80.6 | 87.7 | 86.1 | 85.5 | 90.0 | 80.6 | 88.3 | 86.1 | 85.8 | 90.5 | 80.0 | 88.0 | 36.4 |
| JointFormer (Zhang et al., 2023) | - | - | - | - | - | - | 87.0 | 83.4 | 90.6 | 86.0 | 86.0 | 91.0 | 79.5 | 87.5 | 86.2 | 85.7 | 90.5 | 80.4 | 88.2 | 3.0 |
| JointFormer (Zhang et al., 2023)* | - | - | - | - | - | - | 87.6 | 84.2 | 91.1 | 87.0 | 86.2 | 91.0 | 81.4 | 89.3 | 87.0 | 86.1 | 90.6 | 82.0 | 89.5 | 3.0 |
| **S3 (Ours)** | 68.5 | 64.5 | 72.6 | 66.5 | 62.1 | 70.8 | 86.7 | 82.7 | 90.8 | 87.4 | 87.0 | 92.0 | 80.9 | 89.7 | 87.5 | 86.8 | 91.8 | 81.3 | 89.9 | 13.1 |
| *Trained on the MEGA dataset* | | | | | | | | | | | | | | | | | | | | |
| DEVA (Cheng et al., 2023b) | 66.5 | 62.3 | 70.8 | 54.0 | 49.0 | 59.0 | 83.2 | 79.6 | 86.8 | 86.2 | 85.4 | 89.9 | 80.5 | 89.1 | 85.8 | 84.8 | 89.2 | 80.3 | 88.8 | 25.3 |
| Cutie (Cheng et al., 2023a) * | 69.9 | 65.8 | 74.1 | 66.7 | 62.4 | 71.0 | 86.1 | 82.4 | 89.9 | 87.0 | 86.4 | 91.1 | 81.4 | 89.2 | 87.0 | 86.0 | 90.5 | 82.0 | 89.6 | 36.4 |
| **S3 (Ours)** | 74.0 | 69.8 | 78.3 | 73.0 | 68.3 | 77.8 | 87.8 | 84.0 | 91.7 | 88.0 | 87.0 | 91.8 | 82.5 | 90.7 | 88.1 | 87.4 | 92.5 | 81.9 | 90.7 | 13.1 |

Table 4: **Comparasion with SAM 2** (Ravi et al., 2024). It shows that our method performs favorably against SAM 2.

| Dataset | LVOSV2-val | | | LVOS-val | | | DAVIS17-test | | | MOSE-val | | | YouTube-VOS 2019 val | | |
|---|---|---|---|---|---|---|---|---|---|---|---|---|---|---|---|
| Method | $\mathcal{J}\&\mathcal{F}$ | $\mathcal{J}$ | $\mathcal{F}$ | $\mathcal{J}\&\mathcal{F}$ | $\mathcal{J}$ | $\mathcal{F}$ | $\mathcal{J}\&\mathcal{F}$ | $\mathcal{J}$ | $\mathcal{F}$ | $\mathcal{G}$ | $\mathcal{J}$ | $\mathcal{F}$ | $\mathcal{G}$ | $\mathcal{J}$ | $\mathcal{F}$ |
| SAM 2 (Hiera-B+) | 75.8 | 72.0 | 79.6 | 74.9 | 70.2 | 79.6 | 88.3 | 85.0 | 91.5 | 75.8 | 71.8 | 79.9 | 88.4 | 85.2 | 91.6 |
| SAM 2 (Hiera-L) | 78.1 | 74.3 | 81.9 | 76.1 | 71.6 | 80.6 | 89.0 | 85.8 | 92.2 | 77.2 | 73.3 | 81.2 | 89.1 | 86.0 | 92.2 |
| S3 (ViTb+) | (+7.9) 83.7 | 80.1 | 87.2 | (+1.1) 76.0 | 71.2 | 80.8 | (+1.5) 89.8 | 86.4 | 93.1 | (+0.1) 75.9 | 71.8 | 79.8 | (+0.3) 88.7 | 85.5 | 91.9 |

representative target information. Additionally, the feature extraction module provides matching with more semantic and spatially informative features.

**LVOS** (Hong et al., 2023) is a new long-term dataset to validate the robustness of VOS methods when faced with these challenges. It contains videos 20 times longer than existing VOS datasets, typically featuring a single target per video. In this benchmark, our method achieves a performance gain (**66.5%** compared to **56.2%**). This improvement is attributed to the quality of memorized features and queries, which enhances the robustness of our approach in long-term video scenarios. Our method demonstrates enhanced stability and effectiveness in long-duration contexts by incorporating more discriminative queries that capture crucial target information into the object memory.

**Comparison with Segment Anything 2** (Ravi et al., 2024). Table 4 shows the detailed comparison results between our method and SAM2. Specifically, SAM2 used a large-scale mixed training dataset (comprising 1. 3% DAVIS, approximately 9.4% MOSE, approximately 9.2% YouTubeVOS, approximately 15.5% SA-1B, approximately 49.5% SA-V, and approximately 15. 1% Internal, approximately **120.6k** videos). In contrast, our model is trained solely on VOS datasets (DAVIS, MOSE, YouTubeVOS, OVIS, and BURST, about **11.4K** videos). To ensure a relatively fair comparison, we use the same model size (ViTb) and the same input image size (1024p). In all benchmarks evaluated, our method outperforms SAM2 in all metrics, especially on the challenging LVOSV2 val dataset (**+7.9%**). Even when SAM2 (Ravi et al., 2024) uses Hiera-L as a feature extractor, our model demonstrates comparable results, especially on LVOS V2 (Hong et al., 2023) val dataset (**83.7%** vs **78.1%**).

**Visualized Results.** We visualize some challenging sequences, including scenarios involving part-to-whole-changing and faint objects and those where targets closely resemble the background. In Figure 4(a), our approach shows favorable segmentation performance against the state-of-the-art methods. For example, our model accurately segments targets and provides detailed results in a video that includes gorillas, necks, and a riding man. This can be attributed to our proposed discriminative query propagation, which enables our model to capture more detailed target information by focusing on the most representative features, thereby facilitating precise segmentation in these challenging scenarios Additionally, incorporating semantic features in our model ensures the model understands

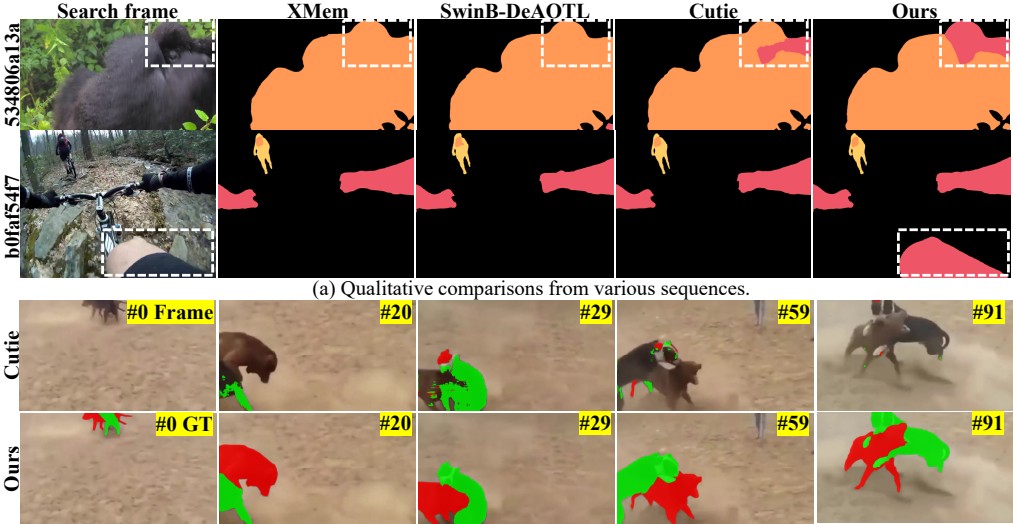

(a) Qualitative comparisons from various sequences.

(b) Qualitative comparisons with target appearance variance.

Figure 4: **Visualized results on challenging scenarios.** The proposed method performs well on these challenging sequences, while other methods suffer from semantic understanding and thus cannot segment the whole objects. (b) Our method handles the challenging sequence with fast-motion objects better than Cutie (Cheng et al., 2023a) .

the semantic prior of the interested targets, such as in the case of the sequence of man riding from YoutubeVOS 2019.

Figure 4(b) shows the segmentation and tracking performance on long-term sequences with fast motion and dramatic appearance variations. In this sequence, two similar dogs continuously change their position and posture, making it challenging for the model to identify and track them accurately. Our VOS method performs accurately in tracking and segmenting targets, whereas Cutie fails to identify and segment the targets in this scenario. Good performance in scenarios like this is due to the semantic-aware feature and discriminative query representation learned by our model. More visualization and videos can be found in the Supplementary Material.

## 4.4 LIMITATIONS

Our method enhances target representation by introducing spatial-semantic features and discriminative query propagation, significantly improving numerous challenging scenarios. However, in scenes with object parts as the target (some sequences in the VOT challenge, shown in the appendix), the proposed model finds it difficult to represent a part of the target and often extends the predicted mask to the whole target. For example, when the target is the head of a person, our model may gradually shift to segmenting the entire body. This tendency arises because our method tends to interpret the target as the entire object seen during the training phase. A potential direction to improve this issue could be to let the model learn more part-aware semantic features, which also needs a new benchmark. Moreover, the improvement of the inference speed will also be addressed in our future work.

## 5 CONCLUSION

In this paper, we propose a spatial-semantic network to learn comprehensive representations to improve the robustness of video object segmentation in handling objects with complex structures or separate parts. In addition, we develop a discriminative query propagation module that selects representative features from target objects for query update, enabling more reliable query propagation in long-term videos. Our approach achieves favorable performance compared with state-of-the-art methods on numerous datasets. Both quantitative and qualitative results demonstrate the effectiveness of the proposed method in tracking and segmenting video objects in complex and long-term scenes.

**Acknowledgments** This work is supported by the Guangdong Basic and Applied Basic Research Foundation (Grant No. 2024A1515011292), the National Natural Science Foundation of China (Grant No.62476148, 62172126), and the Shenzhen Research Council (No. JCYJ20210324120202006).

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

# A APPENDIX

The appendix is organized as follows:

1. We provide a detailed implementation of training and inference.

2. We provide more experiments with different training settings and test hyperparameters.

3. More visualization and failure cases are shown.

## A.1 IMPLEMENT DETAILS.

### A.1.1 MULTI-OBJECT PROCESSING

In this section, we provide additional implementation details for completeness. Our training and inference code will be released to ensure reproducibility.

Following the previous work (Cheng & Schwing, 2022; Cheng et al., 2021a), we extend our methods to the multi-object setting. The predicted mask of each frame undergoes channel separation for different targets. The separated masks for each target are then concatenated with the RGB image and the aggregated mask of other targets for feature extraction. The extracted features are fused with RGB features to obtain pixel-level value features, which are then stored in memory. It is important to note that the pixel-level features stored in memory include the key (RGB feature) and the value.

### A.1.2 TRAINING DETAILS

As mentioned in the main paper, we train our network only on video-level datasets, which differs from other methods (Cheng et al., 2023a; Cheng & Schwing, 2022; Yang & Yang, 2022). The backbone weights are initialized from MAE pre-training, similar to the approaches used in SimVOS (Wu et al., 2023) and JointFormer (Mao et al., 2021). We perform spatial-semantic fusion every three ViT layers and five discriminative query transformer blocks for query propagation. We implement our network using PyTorch and train the model on 8 NVIDIA V100s, employing automatic mixed precision (AMP) during training.

Two different training datasets are used for better improvements. The "DAVIS&YTVOS" settings combine the training sets of YouTubeVOS-2019 and DAVIS-2017, with the DAVIS-2017 dataset expanded five times to increase the data ratio. The "MEGA" set contains four different training sets, including DAVIS-2017, YouYubeVOS-2019, MOSE, and BURST datasets, with the DAVIS-2017 dataset expanded five times to increase the data ratio. To sample a training sequence, we randomly select a "seed" frame from all the frames and then randomly choose 7 other frames from the same video. We re-sample if any two consecutive frames have a temporal distance greater than the max-skip. Following the previous work (Cheng & Schwing, 2022; Cheng et al., 2023a), the max-skip is set to [5, 10, 15, 5] after [0%, 10%, 30%, 80%] of the training iterations, respectively. To avoid sampling frames without targets, we use the data statistics provided by Cutie (Cheng et al., 2023a).

Table 5: **Training parameters.**

| Config | DAVIS YTVOS | MEGA |
|---|---|---|
| optimizer | AdamW | AdamW |
| base learning rate | 5e-5 | 5e-5 |
| weight decay | 0.05 | 0.05 |
| droppath rate | 0.15 | 0.15 |
| batch size | 16 | 16 |
| num ref frames | 3 | 3 |
| num frames | 8 | 8 |
| max-skip | [5, 10, 15, 5] | [5, 10, 15, 5] |
| max-skip-itr | [0.1,0.3,0.8,1] | [0.1, 0.3, 0.8, 1] |
| Iterations | 150,000 | 190,000 |
| learning rate schedule | steplr | steplr |

Table 6: **Quantitative comparisons with high-resolution input on the MOSE, LVOS, DAVIS 2017, YouTube-VOS 2018 & 2019 dataset. + means improve the input size.**

| Dataset / Method | MOSE-val | | | DAVIS 2017 test | | | YouTube-VOS 2018 val | | | | | YouTube-VOS 2019 val | | | | |
|---|---|---|---|---|---|---|---|---|---|---|---|---|---|---|---|---|
| | $\mathcal{J}\&\mathcal{F}$ | $\mathcal{J}$ | $\mathcal{F}$ | $\mathcal{J}\&\mathcal{F}$ | $\mathcal{J}$ | $\mathcal{F}$ | $\mathcal{G}$ | $\mathcal{J}_s$ | $\mathcal{F}_s$ | $\mathcal{J}_u$ | $\mathcal{F}_u$ | $\mathcal{G}$ | $\mathcal{J}_s$ | $\mathcal{F}_s$ | $\mathcal{J}_u$ | $\mathcal{F}_u$ |
| Cutie-base (Cheng et al., 2023a)* | 64.0 | 60.0 | 67.9 | 84.2 | 80.6 | 87.7 | 86.1 | 85.8 | 90.5 | 80.0 | 88.0 | 86.1 | 85.5 | 90.0 | 80.6 | 88.3 |
| ISVOS (Wang et al., 2023)*+BL30K (Cheng et al., 2021b) | - | - | - | 84.0 | 80.1 | 87.8 | 86.7 | 86.1 | 90.8 | 81.0 | 89.0 | 86.3 | 85.2 | 89.7 | 81.0 | 89.1 |
| JointFormer (Zhang et al., 2023)*+BL30K (Cheng et al., 2021b) | - | - | - | 88.1 | 84.7 | 91.6 | 87.6 | 86.4 | 91.0 | 82.2 | 90.7 | 87.4 | 86.5 | 90.9 | 82.0 | 90.3 |
| Ours | 68.5 | 64.5 | 72.6 | 87.1 | 83.1 | 91.1 | 87.4 | 87.0 | 92.0 | 80.9 | 89.7 | 87.5 | 86.8 | 91.8 | 81.3 | 89.9 |
| Cutie-base (Cheng et al., 2023a)*+ | 66.2 | 62.3 | 70.1 | 85.9 | 82.6 | 89.2 | - | - | - | - | - | 86.9 | 86.2 | 90.7 | 81.6 | 89.2 |
| Ours+ | 70.5 | 66.5 | 74.6 | 87.9 | 84.6 | 91.3 | 87.6 | 86.9 | 91.7 | 81.5 | 90.1 | 87.8 | 86.8 | 91.6 | 82.2 | 90.8 |
| Cutie-base* (Cheng et al., 2023a) w/ MEGA | 69.9 | 65.8 | 74.1 | 86.1 | 82.4 | 89.9 | 87.0 | 86.4 | 91.1 | 81.4 | 89.2 | 87.0 | 86.0 | 90.5 | 82.0 | 89.6 |
| Ours w/MEGA | 73.2 | 68.8 | 77.5 | 88.2 | 84.3 | 92.1 | 88.1 | 87.4 | 92.5 | 81.9 | 90.7 | 88.0 | 88.0 | 91.8 | 82.5 | 90.8 |
| Cutie-base (Cheng et al., 2023a)*+ w/ MEGA | 71.7 | 67.6 | 75.8 | 88.1 | 84.7 | 91.4 | - | - | - | - | - | 87.5 | 86.3 | 90.6 | 82.7 | 90.5 |
| Ours+ w/MEGA | **75.1** | **71.0** | **79.2** | **89.1** | **85.8** | **92.4** | **88.5** | **87.6** | **92.6** | **82.7** | **91.3** | **88.5** | **87.3** | **92.0** | **83.1** | **91.4** |

We conduct data augmentation with methods of random horizontal mirroring, random affine transformations, and cut-and-paste. The training samples are also resized by crop (480 x 480). Then, random color jittering and random grayscaling are performed in the training samples. The same augmentation is performed in the same video for stability. To ensure the images can be input into the patch embedding of ViT, we apply padding to both the images and masks so that their dimensions are multiples of 32.

We initialize the model with the first frame and ground truth in the training phase. When the number of past frames is three or fewer, we use all of them as memory frames. If there are more than three past frames, we randomly sample three of them to be the memory frames. We compute the loss at all frames except the first one and perform back-propagation through time.

### A.1.3 INFERENCE DETAILS

During the inference phase, the model memorizes the feature every 5th frame like (Cheng & Schwing, 2022; Cheng et al., 2023a). The pixel-level memory always stores the first frame and its mask. The top-K filter is used to augment memory reading in pixel memory. We use the streaming average for object-level memory to accumulate the discriminative query representation like the operation in (Cheng et al., 2023a).

## A.2 ADDITIONAL QUANTITATIVE RESULTS

In this section, we provide additional ablation studies and SOTA comparisons with different settings.

Table 6 shows detailed results about methods with further pre-training (e.g. JointFormer (Zhang et al., 2023) and ISVOS (Wang et al., 2023)) and improved test size (Cutie (Cheng et al., 2023a))We note that our method achieves comparable results against SOTA methods (Cutie (Cheng et al., 2023a) and JointFormer (Zhang et al., 2023)) without any pre-training. When trained with MEGA datasets or improved test image size, our method achieves state-of-the-art results in all the benchmarks.

In the main paper, we provide ablation studies for different components. To provide a broader comparison, we conduct tests using various combinations of training data and testing settings. The experimental results in Table 7 detail the impact of different components of our model on the performance, facilitating a better understanding of the effectiveness of our model.

To compare with more methods on MOSE, we follow the training strategy from MOSE (Ding et al., 2023), and replace the YouTubeVOS with MOSE in the training process. Table 8 shows the results of the variants of our method. All the variants achieve great improvement in terms of $\mathcal{J}\&\mathcal{F}$, which further validates the effectiveness of the proposed spatial-semantic feature learning and discriminative query representation.

Table 9 shows the results of our method on the more challenging VOTS2023 (Kristan et al., 2023) benchmark, which indicates that our method outperforms the winner (DMAOT) of last year on most metrics. Our method achieves the highest robustness thanks to our representation of target saliency information and the modeling of target semantic-spatial information.

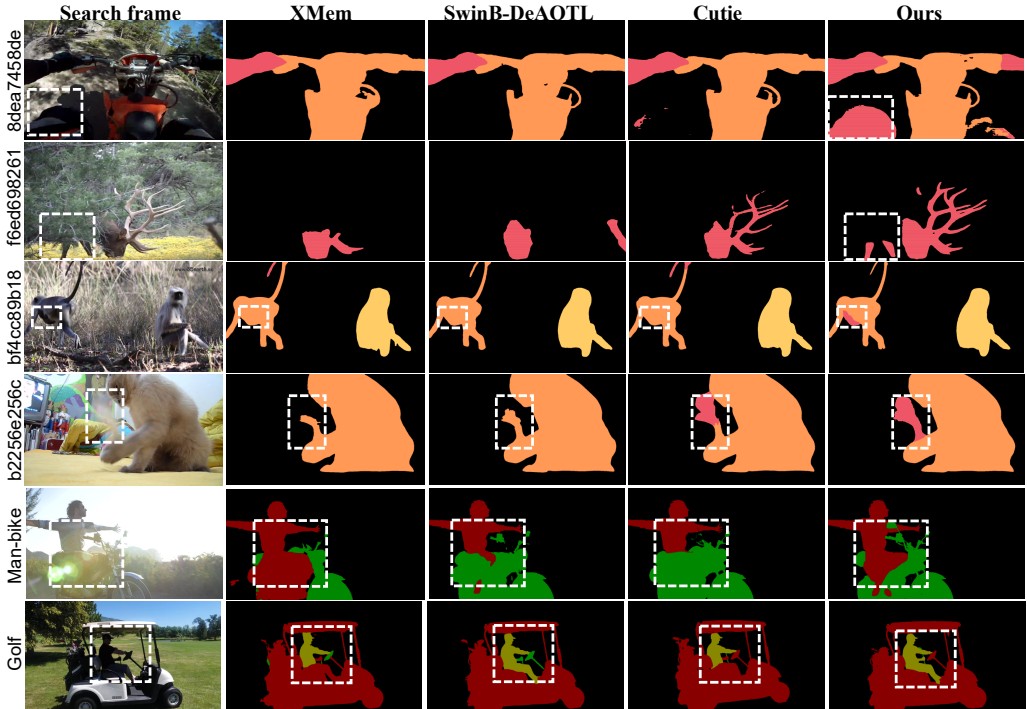

Qualitative comparisons from various sequences.

Figure 5: **Additional qualitative comparison against state-of-the-art methods including XMem, Swin-DeAOT, and Cutie.** All the sequences are selected from YouTubeVOS 2019 and DAVIS 2017 datasets.

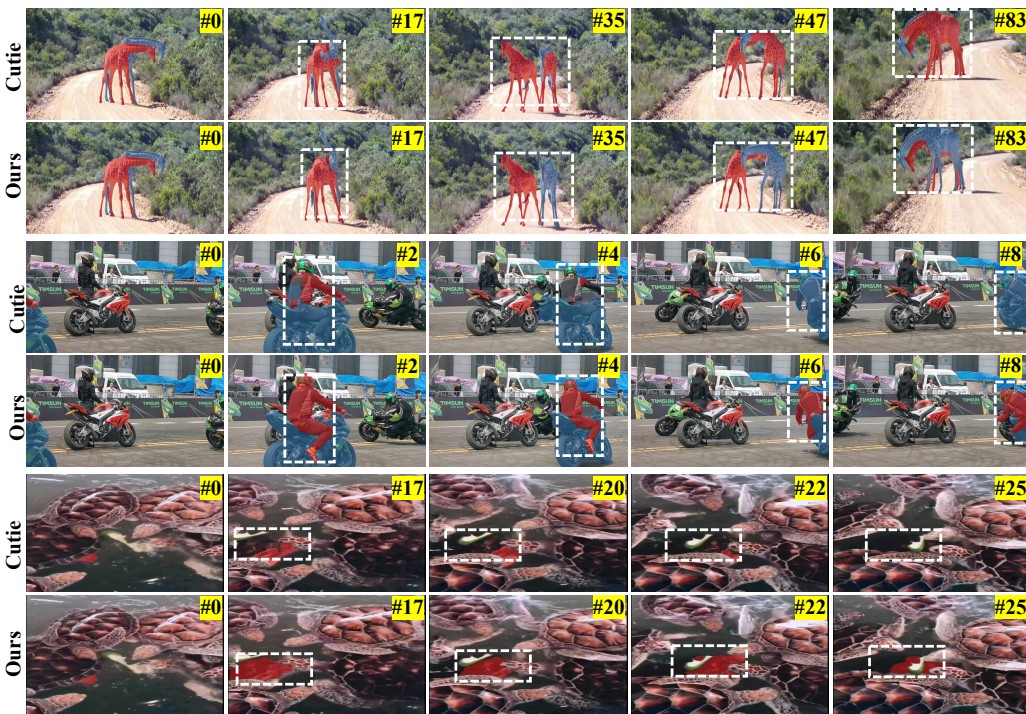

Qualitative comparisons from various sequences with appearance variance.

Figure 6: **Additional qualitative comparison on sequence with giant appearance variance.** All the sequences are picked from MOSE (Ding et al., 2023) dataset.

Table 7: **Model detailed ablation study.**

| Dataset | MOSE-val | | | DAVIS 2017 test | | | LVOS test | | | YouTube-VOS 2019 val | | | | |
|---|---|---|---|---|---|---|---|---|---|---|---|---|---|---|
| Method | $\mathcal{J}\&\mathcal{F}$ | $\mathcal{J}$ | $\mathcal{F}$ | $\mathcal{J}\&\mathcal{F}$ | $\mathcal{J}$ | $\mathcal{F}$ | $\mathcal{G}$ | $\mathcal{J}$ | $\mathcal{F}$ | $\mathcal{G}$ | $\mathcal{J}_s$ | $\mathcal{F}_s$ | $\mathcal{J}_u$ | $\mathcal{F}_u$ |
| *Trained on the YouTubeVOS, and DAVIS datasets* | | | | | | | | | | | | | | |
| XMem (Cheng & Schwing, 2022) (Baseline) | 53.3 | 62.0 | 57.6 | 81.0 | 77.4 | 84.5 | 50.0 | 45.5 | 54.4 | 85.5 | 84.3 | 88.6 | 80.3 | 88.6 |
| Cutie (Cheng et al., 2023a) | 64.0 | 60.0 | 67.9 | 84.2 | 80.6 | 87.7 | 56.2 | 51.8 | 60.5 | 86.1 | 85.8 | 90.5 | 80.0 | 88.0 |
| +Discriminative Query | 64.2 | 60.3 | 68.1 | 85.2 | 81.8 | 88.5 | 57.4 | 53.3 | 61.5 | 86.5 | 86.2 | 90.7 | 80.6 | 88.8 |
| +ViT | 64.2 | 60.2 | 68.3 | 85.6 | 82.0 | 89.2 | 58.3 | 53.7 | 62.8 | 86.7 | 86.4 | 90.3 | 81.0 | 88.7 |
| +Spatial | 68.2 | 64.0 | 72.4 | 86.2 | 82.4 | 90.1 | 67.4 | 62.9 | 71.9 | 87.3 | 86.4 | 91.3 | 81.5 | 90.3 |
| +Semantic (Full) | 68.5 | 64.5 | 72.6 | 86.7 | 82.7 | 90.8 | 66.5 | 62.1 | 70.8 | 87.5 | 86.8 | 91.8 | 81.3 | 89.9 |
| *Improved test size (600/720)* | | | | | | | | | | | | | | |
| Cutie (Cheng et al., 2023a) | 66.2 | 62.3 | 70.1 | 85.9 | 82.6 | 89.2 | 56.2 | 51.8 | 60.5 | 86.9 | 86.2 | 90.7 | 81.6 | 89.2 |
| +Discriminative Query | 66.4 | 62.4 | 70.1 | 87.9 | 84.6 | 91.2 | 57.4 | 53.3 | 61.5 | 87.1 | 86.3 | 90.6 | 82.0 | 89.5 |
| +Spatial | 69.9 | 67.5 | 74.1 | 87.0 | 83.7 | 90.2 | 67.4 | 62.9 | 71.9 | 87.5 | 86.8 | 91.6 | 81.6 | 90.2 |
| +Semantic ( full) | 70.5 | 66.5 | 74.6 | 87.8 | 84.6 | 91.3 | 66.5 | 62.1 | 70.8 | 87.7 | 86.8 | 91.6 | 82.2 | 90.8 |
| *Trained on the MEGA datasets* | | | | | | | | | | | | | | |
| Cutie (Cheng et al., 2023a) | 69.9 | 65.8 | 74.1 | 86.1 | 82.4 | 89.9 | 66.7 | 62.4 | 71.0 | 87.0 | 86.0 | 90.5 | 82.0 | 90.7 |
| +Discriminative query | 70.6 | 66.5 | 74.6 | 86.6 | 82.7 | 90.5 | 66.5 | 62.1 | 70.8 | 87.5 | 86.0 | 90.6 | 82.8 | 90.6 |
| +Spatial | 73.5 | 69.1 | 77.7 | 87.6 | 83.8 | 91.5 | 68.8 | 64.4 | 73.1 | 87.9 | 86.9 | 91.8 | 82.3 | 90.4 |
| +Semantic (Full) | 74.0 | 69.8 | 78.3 | 87.8 | 84.0 | 91.7 | 73.0 | 68.3 | 77.8 | 88.1 | 87.4 | 92.5 | 81.9 | 90.7 |
| *Improved test size (600/720)* | | | | | | | | | | | | | | |
| Cutie (Cheng et al., 2023a) | 71.7 | 67.6 | 75.8 | 88.1 | 84.7 | 91.4 | 66.7 | 62.4 | 71.0 | 87.5 | 86.3 | 90.6 | 82.7 | 90.5 |
| +Discriminative query | 71.6 | 67.7 | 75.5 | 88.1 | 84.6 | 91.4 | 66.5 | 62.1 | 70.8 | 88.0 | 86.2 | 90.6 | 83.6 | 91.5 |
| +Spatial | 75.3 | 71.3 | 79.2 | 89.0 | 85.8 | 92.3 | 68.8 | 64.4 | 73.1 | 88.3 | 87.0 | 91.9 | 83.0 | 91.1 |
| +Semantic(Full) | 75.1 | 71.0 | 79.2 | 89.1 | 85.8 | 92.4 | 73.0 | 68.3 | 77.8 | 88.5 | 87.3 | 92.0 | 83.1 | 91.4 |

Table 8: **Results only trained on MOSE datasets.**

| Dataset | MOSE-val | | |
|---|---|---|---|
| Method | $\mathcal{J}\&\mathcal{F}$ | $\mathcal{J}$ | $\mathcal{F}$ |
| RDE (Li et al., 2022a) | 48.8 | 44.6 | 52.9 |
| STCN (Cheng et al., 2021a) | 50.8 | 46.6 | 55.0 |
| AOT (Yang et al., 2021) | 57.2 | 53.1 | 61.3 |
| XMem (Cheng & Schwing, 2022) | 57.6 | 53.3 | 62.0 |
| DeAOT (Yang & Yang, 2022) | 59.4 | 55.1 | 63.8 |
| ResNet+Discriminative query | 69.9 | 65.8 | 73.9 |
| +Spatial | 72.7 | 68.3 | 77.0 |
| +Semantic | 72.9 | 68.4 | 77.3 |
| *Improved test size (720)* | | | |
| ResNet+Discriminative query | 71.6 | 67.7 | 75.5 |
| +Spatial | 74.0 | 69.9 | 78.1 |
| +Semantic | 74.5 | 70.2 | 78.5 |

Table 9: **Results on VOTS2023 Challenge.**

| Dataset | VOTS2023 | | | | | |
|---|---|---|---|---|---|---|
| Method | $Q\uparrow$ | $A\uparrow$ | $R\uparrow$ | $NER\downarrow$ | $DRE\downarrow$ | $ADQ\downarrow$ |
| Dynamic_DEAOT | 0.59 | 0.69 | 0.84 | 0.07 | 0.1 | 0.57 |
| M-VOSTrack | 0.61 | 0.75 | 0.76 | 0.16 | 0.08 | 0.71 |
| HQTrack | 0.62 | 0.75 | 0.77 | 0.16 | 0.08 | 0.70 |
| DMAOT | 0.64 | **0.75** | 0.80 | 0.14 | 0.07 | 0.73 |
| Ours | **0.67 (+0.3)** | 0.74 | **0.86 (+0.6)** | **0.09 (+0.5)** | **0.05 (+0.02)** | **0.75 (+0.02)** |

## A.3 ADDITIONAL QUALITATIVE COMPARISON

We present a more qualitative comparison of most benchmarks in Figure 5 and Figure 6.

In Figure 5, we categorize the scenarios into three main types: overall-to-part segmentation (man on bike and deer), small object segmentation (small monkey), blurry object segmentation (cat toy), and segmentation under varying lighting conditions (man-bike and golf). In these three cases, our method demonstrates precise segmentation compared to other methods. The semantic embedding allows our model to consider semantic information when parsing objects, spatial dependence modeling enhances the understanding of spatial positions and details of the object, and the discriminative query

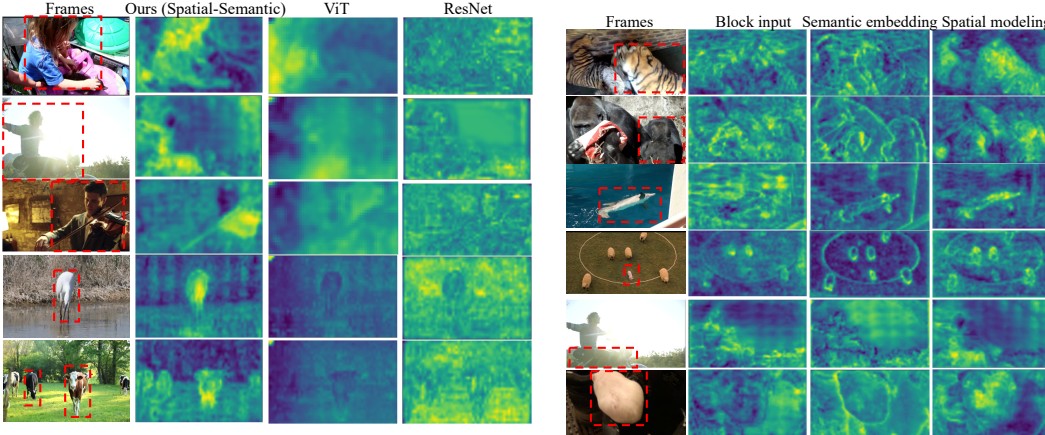

(a) Visualization of the feature map output from different backbones

(b) Visualization of the feature map after proposed semantic and spatial layers

Figure 7: **Visualizations of feature maps from different architectural components**.

accurately represents the target. The integration of these three components enables the model to achieve excellent segmentation and tracking performance in complex and challenging scenarios.

In Figure 6, we select more sequence results on MOSE, primarily divided into two categories: positional changes of similar targets and rapid movement of targets. In both cases, our method outperforms Cutie. This is attributed to our proposed spatial-semantic feature representation and discriminative query expression, which enable the model to distinguish appearance-similar targets and consistently track the target.

Figure 9 compares sequences that require a semantic understanding of the target. Without the semantic embedding module, the method suffers from segmenting parts of the target. The addition of the semantic embedding module enables the model to fully understand the semantic information of the target, allowing for precise segmentation and tracking in these scenarios.

**Visualization of feature maps**. Figure 7(a) shows that the proposed Spatial-Semantic block assigns higher activation weights to complex target objects compared to the CNN or ViT model, which indicates that the proposed model makes target representations more distinguishable than ViT and ResNet. Moreover, we show how the feature changes during the processing of the proposed model by visualizing the feature maps at different processing steps in 7(b). After passing through the semantic embedding module, the feature maps (semantic embedding) demonstrate that the module makes the contours of objects clearer, indicating richer semantic information. After passing through the spatial dependence module, the feature maps (spatial modeling) show that the details of the targets are more pronounced. The changes in the feature maps confirm that the proposed two modules significantly enhance the semantic and spatial information of the targets.

We also visualize the similarity between the learned discriminative queries and the feature map using heat maps in Figure 8. The visualization shows that the learned discriminative queries distinctly express the target areas. We also mark the positions of the discriminative queries, which represent the locations of the pixels with the highest similarity.

**Failure cases** are shown in Figure 10, in which our method tends to segment the whole instance rather than the part of it. The challenging sequences require the model to segment and track the head of the man, the flamingo, and the kangaroo. Our model interprets semantics as the entire instance during tracking and segmentation, such as a person, bird, or kangaroo, resulting in segmentation outcomes gradually favoring the whole instance.

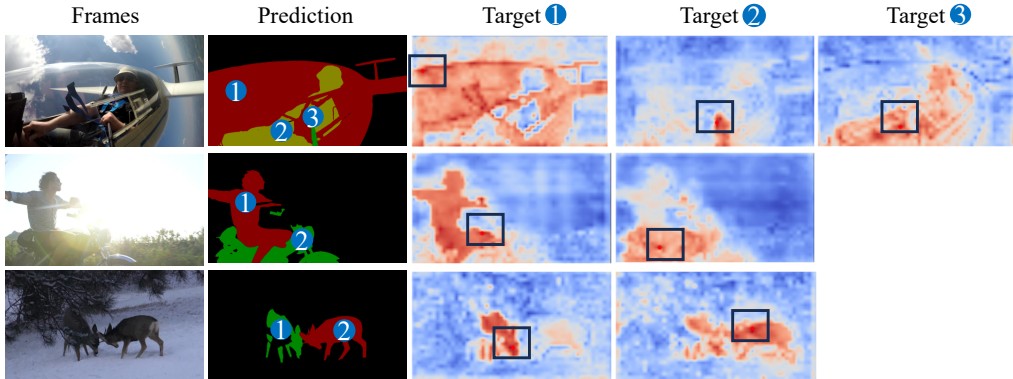

Figure 8: **Visualization of the similarity map between query and features**. The map indicates the discrimination of the target representation. The discriminative queries of different targets are plotted in the similarity map, which can be found in the black box.

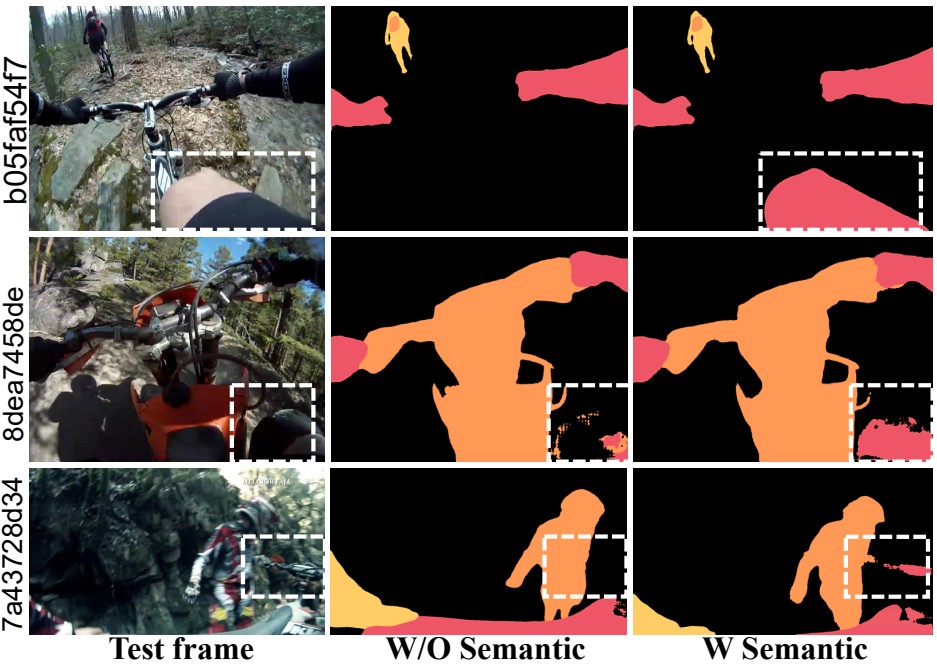

Figure 9: **Qualitative comparison between with and without semantic embedding block.** All the sequences are picked from the YouTubeVOS 2019 dataset.

**Time line**

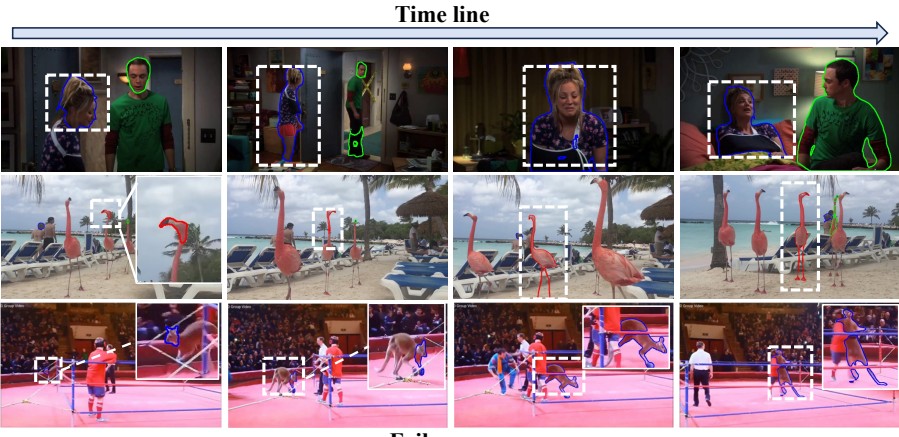

**Failure cases**

Figure 10: **Failure cases.** All the sequences are picked from VOT Challenge (Kristan et al., 2023).

