# OpenReview forum: "Learning Spatial-Semantic Features for Robust Video Object Segmentation"
_ICLR.cc/2025/Conference — ICLR 2025 Poster_

### Official Review · Reviewer_eHC5 · 2024-10-28

**Soundness:** 4
**Presentation:** 4
**Contribution:** 4
**Rating:** 8
**Confidence:** 5

**Summary:**

The authors focus on the problem of video object segmentation in long-term tracking and complex environments. To improve the model's robustness, a spatial-semantic network block is proposed to integrate semantic information with spatial information for video object segmentation. Additionally, a discriminative query mechanism is developed to capture the most representative region of the target for better target representation learning and updating. The proposed method achieves state-of-the-art results on most VOS dataset.

**Strengths:**

This paper addresses key issues in current applications of VOS methods: long-term tracking, occlusion, and object representation changes.
1.	The proposed approach utilizes semantic and spatial information from upstream pre-trained models, enriching the target's semantic and detailed information. It is novel in the field of VOS.
2.	The paper proposes a discriminative query generation mechanism to provide the model with more distinctive target information, which is validated on LVOS datasets.
3.	The proposed method is validated on various VOS datasets and achieves state-of-the-art results.

**Weaknesses:**

The paper does not have obvious weaknesses, but there are still some issues.
1.	In Table 1, why is there no separate ablation study for the spatial block and semantic block? Please provide this part of the experiment.
2.	Some detail issues: In Figure 3, the blue results are not clearly marked and the position of * is not aligned.

**Questions:**

1、	How many trainable parameters and total parameters does the model have?
2、	Why does "DepthAnything" achieve the best results?
3、	If the backbone of the Cutie model is replaced with ViT, would it achieve good results?

---

> ### Author Response · Authors · 2024-11-19
> **Response to Reviewer eHC5**
>
> We sincerely thank Reviewer eHC5 for reviewing this paper.
>
> **Q1\: In Table 1, why is there no separate ablation study for the spatial block and semantic block\?**
> &nbsp;&nbsp; To save space, we place the detailed ablation study in Appendix Table 7. Table 7 shows that spatial dependence modeling significantly improves performance across all VOS datasets, with a **3.0%+ J&F gain on MOSE val**. This improvement comes from better detail modeling during feature extraction. Adding semantic information further boosts performance, reaching **73% J&F on the LVOS test**, demonstrating its value in enhancing target semantic understanding. We provide some experiments in the table below. For more details, please refer to the Appendix.
>
> | **Dataset**                        | **MOSE** | **D17 test** | **LVOS test** | **YT19** |
> |------------------------------------|----------|--------------|---------------|----------|
> | **Method**                         | **J&F**  | **J&F**      | **J&F**       | **J&F**  |
> | **Trained on YouTubeVOS and DAVIS datasets** |          |              |               |          |
> | XMem (Baseline)                    | 53.3     | 81.0         | 50.0          | 85.5     |
> | Cutie                              | 64.0     | 84.2         | 56.2          | 86.1     |
> | +Discriminative Query               | 64.2     | 85.2         | 57.4          | 86.5     |
> | +Spatial                            | 68.2     | 86.2         | 67.4          | 87.3     |
> | +Semantic (Full)                    | 68.5     | 86.7         | 66.5          | 87.5     |
> | **Trained on the MEGA datasets**   |          |              |               |          |
> | Cutie                              | 69.9     | 86.1         | 66.7          | 87.0     |
> | +Discriminative Query               | 70.6     | 86.6         | 66.5          | 87.5     |
> | +Spatial                            | 73.5     | 87.6         | 68.8          | 87.9     |
> | +Semantic (Full)                    | 74.0     | 87.8         | 73.0          | 88.1     |
>
>
> **Q2\: Some detail issues: In Figure 3, the blue results are not clearly marked and the position of * is not aligned.**
> &nbsp;&nbsp; We have fixed these details in the revised version.
>
> **Q3\:  How many trainable parameters and total parameters does the model have\?**
> &nbsp;&nbsp; In our full version, the total parameters are **226.8 million** and the trainable parameters are **54.1 million**. The trainable parameters are mainly in the Spatial-Semantic Blocks, the target association module, and the decoder.
>
> **Q4\: Why does "DepthAnything" achieve the best results\?**
> &nbsp;&nbsp; DepthAnything provides more powerful representation with pixel-level and depth information compared to other pretrained models, which enables efficient target feature representation across diverse scenarios. This is also validated on instance segmentation in the DepthAnything paper.
>
> **Q5:  If the backbone of the Cutie model is replaced with ViT, would it achieve good results?**
> &nbsp;&nbsp; The table below compares Cutie with different backbones and our proposed model. The results show that replacing Cutie's backbone with ViT improves performance slightly (**\+0.3% on YT19 and 0.2% on MOSE**). This is because ViT provides stronger representation capabilities compared to ResNet. However, simply replacing the backbone with ViT fails to produce multi-scale and the associated semantic information. To address this issue, the proposed Spatial-Semantic Block jointly models semantic and multi-scale spatial information, achieving notable improvements across different datasets (**\+1.4% on YT19 and 4.5% on MOSE**).
> | Methods | Backbone  | D17 test (J&F) | YT19 (J&F) | MOSE (J&F) |
> |---------|-----------|----------------|------------|------------|
> | Cutie   | ResNet50  | 84.2           | 86.1       | 64.0       |
> | Cutie   | ViTb      | 84.7           | 86.4       | 64.2       |
> | Ours    | ViTb      | 86.7           | 87.5       | 68.5       |

---

> ### Author Response · Authors · 2024-11-23
> **Please let us know if we address all the issues**
>
> Dear Reviewer,
>
> Thank you for the comments on our paper. We have provided a response and a revised paper on Openreview. Since the discussion phase ends on Nov 26, we would like to know whether we have addressed all the issues, and we look forward to resolving any additional questions or concerns you may have.
>
> Thank you again for your time and effort.
>
> Best regards

---

> > ### Comment · Reviewer_eHC5 · 2024-11-25
> >
> > Thanks for your response. My concerns have been solved, and I have adjusted my score. Good luck.

---

> > > ### Author Response · Authors · 2024-11-26
> > > **Thanks a lot**
> > >
> > > Dear Reviewer,
> > >
> > > We sincerely appreciate your acknowledgment and dedication in reviewing our work.
> > >
> > > Thank you

---

### Official Review · Reviewer_ZJce · 2024-11-02

**Soundness:** 3
**Presentation:** 3
**Contribution:** 2
**Rating:** 6
**Confidence:** 4

**Summary:**

This paper presents a novel spatial-semantic block that effectively integrates semantic information with spatial features, resulting in a more comprehensive representation of target objects, especially those with complex or distinct parts. By utilizing a pre-trained Vision Transformer (ViT) backbone without the need to retrain all parameters, the proposed method significantly enhances the efficiency of video object segmentation (VOS).

Additionally, the development of a discriminative query mechanism marks a substantial advancement in the field. This mechanism prioritizes the most representative regions of target objects, thereby improving the reliability of target representation and query updates. This is particularly advantageous in long-term video scenarios, where appearance changes and occlusions can lead to noise accumulation during query propagation.

The authors also highlight the importance of learning comprehensive target features that encompass semantic, spatial, and discriminative information. This holistic approach effectively addresses challenges related to appearance variations and identity confusion among similar-looking objects in long-term videos, making it a valuable contribution to the VOS community.

Finally, extensive experimental results demonstrate that the proposed method achieves state-of-the-art performance across multiple benchmark datasets, including DAVIS 2017, YouTube VOS 2019, MOSE, and LVOS.

**Strengths:**

This paper presents a spatial-semantic modeling method and a discriminative query mechanism that significantly enhance the model's performance. Extensive experiments have been conducted to demonstrate the effectiveness of the model, and several visual examples are provided to clearly illustrate the results at different processing stages. Additionally, the final results showcase the model's considerable potential.

**Weaknesses:**

Writing Style:
1. The writing language is not concise enough, with many long sentences that significantly reduce readability. This is particularly evident in the introduction, such as on the second page: "We construct a Spatial-Semantic Block comprising a semantic embedding module and a spatial dependencies modeling module to efficiently leverage the semantic information and local details of the pre-trained ViTs for VOS without training all the parameters of the ViT backbone."

Image Details:
1. In Figure 2, there are N spatial-semantic blocks, but N is not specified later in the paper.

Method:
1. In Figure 2, the argmax operation in the distinctive query propagation is non-differentiable. Will this prevent the gradient from being propagated through the model?

2. If the introduced ViT backbone is not fine-tuned, will its performance degrade on the new dataset? A comparison experiment between freezing and not freezing the parameters is needed here.

3. The number of different queries should be related to the number of targets. However, using 8 queries yields better results. When faced with more than 8 targets, can 8 queries adequately represent the different targets?

4. In Table 3, there are two XMem entries, one of which is not referenced. It is unclear what the unreferenced entry represents, and why it lacks FPS results needs to be clarified.

5. Table 3 lacks a comparison of Joint Former results trained on the MEGA dataset. Please provide the results for Joint Former trained on the MEGA dataset in detail. If the original Joint Former was not trained on this dataset, can it be trained and then compared for performance?

6. The spatial-semantic block consists of two parts: first, the global feature cls token is fused with the semantic features, and then further enhanced through Deformable Cross Attention. It is necessary to separately validate the effects of directly fusing the features versus applying Deformable Cross Attention for further enhancement.

**Questions:**

1.In Figure 2, there are N spatial-semantic blocks, but N is not specified later in the paper.

2.In Figure 2, the argmax operation in the distinctive query propagation is non-differentiable. Will this prevent the gradient from being propagated through the model?

3.If the introduced ViT backbone is not fine-tuned, will its performance degrade on the new dataset? A comparison experiment between freezing and not freezing the parameters is needed here.

4.The number of different queries should be related to the number of targets. However, using 8 queries yields better results. When faced with more than 8 targets, can 8 queries adequately represent the different targets?

5.In Table 3, there are two XMem entries, one of which is not referenced. It is unclear what the unreferenced entry represents, and why it lacks FPS results needs to be clarified.

6.Table 3 lacks a comparison of Joint Former results trained on the MEGA dataset. Please provide the results for Joint Former trained on the MEGA dataset in detail. If the original Joint Former was not trained on this dataset, can it be trained and then compared for performance?

7.The spatial-semantic block consists of two parts: first, the global feature cls token is fused with the semantic features, and then further enhanced through Deformable Cross Attention. It is necessary to separately validate the effects of directly fusing the features versus applying Deformable Cross Attention for further enhancement.

---

> ### Author Response · Authors · 2024-11-19
> **Response to Reviewer ZJce**
>
> We sincerely thank Reviewer ZJce for reviewing this paper.
>
> **Q1: Writing**
> &nbsp;&nbsp; We have rewritten the long sentences to shorter ones and polished the manuscript based on the suggestions to make it concise and clear, which can be seen in the revised version.
>
> **Q2: Model Details: N blocks**
> &nbsp;&nbsp; Our model includes 4 spatial-semantic blocks (i.e. N=4), each interacting with the ViTb (12 layers) model every three ViT layers. This implementation detail is included in the Implementation Details section. As mentioned in the manuscript, we will release our source code and model to the public.
>
> **Q3: Argmax operation**
> &nbsp;&nbsp; We use the Argmax operation to select the index with the highest similarity and do not perform back-propagation. The backpropagation of the relevant part is achieved through gradient flow via skip connections.
>
> **Q4: Will the performance on new datasets degrade when the ViT backbone is not fine-tuned\?**
> &nbsp;&nbsp; Our model was trained on the YT+DAVIS dataset and the MEGA dataset. Without any training on the LVOS dataset, it achieved SOTA results directly during testing (**66.5% and 73.0%**). This demonstrates that our model's performance **does not degrade when transferred to new datasets.**
>
> &nbsp;&nbsp; The designed SS block is specifically aimed at better-adapting features from upstream pretrained models to the VOS task. The semantic embedding module integrates semantic information, while spatial dependence modeling facilitates the interaction of multi-scale spatial features. Through this block, we effectively adapt a pretrained model to generate the multi-scale requirements of VOS tasks without the need for fine-tuning.
>
> &nbsp;&nbsp; In our ablation study (**Table 1, Row 4**), we directly replaced the feature extraction module with ViT and used FPN to extract multi-scale features. Although this experiment involved full fine-tuning of ViT, it did not significantly improve performance.
>
> &nbsp;&nbsp; We attempted to perform full fine-tuning of our complete model on NVIDIA A100 GPUs (40GB). However, the training encountered Out-Of-Memory errors, indicating that full fine-tuning requires more computational resources and training time.
>
> **Q5: When faced with more than 8 targets, can 8 queries adequately represent the different targets?**
> &nbsp;&nbsp;  The table below provides query numbers with additional values of 4, 10, and 64. The results show that:
> &nbsp;&nbsp;  (1) Increasing the number of queries does not consistently improve the model's performance. Using larger numbers of queries may degrade the performance slightly, as using too many queries will introduce more noise during training and slow down the convergence speed.
>
> | Queries | D17 (J&F) | MOSE-val (J&F) | YT19 (J&F) | LVOS (J&F) | YT18 (J&F) |
> |---------|-------------|----------------|------------|------------|------------|
> | 64      | 85.0        | 65.5           | 86.7       | 65.2       | 86.6       |
> | 32      | 85.8        | 68.3           | 86.9       | 64.8       | 86.8       |
> | 16      | 86.6        | 67.7           | 87.0       | 66.4       | 86.9       |
> | 10      | 86.4        | 68.2           | 87.0       | 66.4       | 87.1       |
> | **8**   | **86.7**    | **68.5**       | **87.5**   | **66.5**   | **87.4**   |
> | 4       | 83.1        | 60.2           | 85.2       | 53.5       | 85.2       |
>
> &nbsp;&nbsp; (2) The model achieves the best performance when the number of queries is set to 8. This is primarily because the number of queries has a close relationship with the number of targets. The table below presents the target number statistics across different datasets, which are mostly concentrated between 1 and 5. Setting the query number to 8 effectively covers almost all sequences in the datasets, as verified by the experimental results.
>
> | Target numbers | YTVOS19 | MOSE | LVOSv2 |
> |-------------------|---------|------|--------|
> | 1                 | 168     | 188  | 91     |
> | 2                 | 171     | 58   | 31     |
> | 3                 | 132     | 15   | 8      |
> | 4                 | 26      | 22   | 4      |
> | 5                 | 7       | 11   | 0      |
> | 6                 | 3       | 8    | 2      |
> | 7                 | 0       | 4    | 1      |
> | >=8               | 0       | 5    | 2      |
>
> **Q6：Lack reference of the XMem in Table 3.**
> &nbsp;&nbsp; These two XMem settings represent the same method with different configurations, where the one marked with * is pretrained on static images. We have added the citation in the revised manuscript.

---

> ### Author Response · Authors · 2024-11-19
> **Response to Reviewer ZJce (2)**
>
> **Q7: JointFormer trained on MEGA datasets.**
> &nbsp;&nbsp; JointFormer has not released its source code, so we tried our best to reproduce the code based on the paper. Unfortunately, the reproduced results differ from those reported in the original paper, making it impossible for us to compare it with our method on the MEGA dataset. Although it was not possible to train JointFormer on the MEGA dataset, a comparison can still be made using models trained only on the YT+DAVIS dataset. Our model achieves better performance than JointFormer on the large-scale datasets YT19 and YT18 with gains of **1.4%** and **1.3%**, respectively.
>
> **Q8: What is the individual impact of directly fusing features versus applying Deformable Cross Attention for further enhancement in the spatial-semantic block?**
> &nbsp;&nbsp; To save space, we place the detailed ablation study in Appendix Table 7. Table 7 shows that spatial dependence modeling significantly improves performance across all VOS datasets, with a **3.0%+ J&F gain on MOSE val**. This improvement comes from better detail modeling during feature extraction. Adding semantic information further boosts performance, reaching **73% J&F on the LVOS test**, demonstrating its value in enhancing target semantic understanding. We provide some experiments in the table below. For more details, please refer to the Appendix.
>
> | **Dataset**                        | **MOSE** | **D17** | **LVOS test** | **YT19** |
> |------------------------------------|----------|--------------|---------------|----------|
> | **Method**                         | **J&F**  | **J&F**      | **J&F**       | **J&F**  |
> | **Trained on YouTubeVOS and DAVIS datasets** |          |              |               |          |
> | XMem (Baseline)                    | 53.3     | 81.0         | 50.0          | 85.5     |
> | Cutie                              | 64.0     | 84.2         | 56.2          | 86.1     |
> | +Discriminative Query               | 64.2     | 85.2         | 57.4          | 86.5     |
> | +Spatial                            | 68.2     | 86.2         | 67.4          | 87.3     |
> | +Semantic (Full)                    | 68.5     | 86.7         | 66.5          | 87.5     |
> | **Trained on the MEGA datasets**   |          |              |               |          |
> | Cutie                              | 69.9     | 86.1         | 66.7          | 87.0     |
> | +Discriminative Query               | 70.6     | 86.6         | 66.5          | 87.5     |
> | +Spatial                            | 73.5     | 87.6         | 68.8          | 87.9     |
> | +Semantic (Full)                    | 74.0     | 87.8         | 73.0          | 88.1     |

---

> ### Author Response · Authors · 2024-11-23
> **Please let us know if we address all the issues**
>
> Dear Reviewer,
>
> Thank you for the comments on our paper. We have provided a response and a revised paper on Openreview. Since the discussion phase ends on Nov 26, we would like to know whether we have addressed all the issues, and we look forward to resolving any additional questions or concerns you may have.
>
> Thank you again for your time and effort.
>
> Best regards

---

> > ### Comment · Reviewer_ZJce · 2024-11-25
> >
> > Thanks for your response. My concerns have been solved, and I have adjusted my score. Good luck.

---

> > > ### Author Response · Authors · 2024-11-26
> > > **Thanks a lot**
> > >
> > > Dear Reviewer,
> > >
> > > We sincerely appreciate your acknowledgment and dedication in reviewing our work.
> > >
> > > Thank you

---

### Official Review · Reviewer_dBZh · 2024-11-02

**Soundness:** 3
**Presentation:** 3
**Contribution:** 3
**Rating:** 6
**Confidence:** 5

**Summary:**

This paper addresses the complex task of tracking and segmenting multiple similar objects in long-term videos, where identifying target objects becomes challenging due to factors like occlusion, cluttered backgrounds, and appearance changes. To tackle these issues, the authors propose a new framework for robust video object segmentation, focusing on learning spatial-semantic features and generating discriminative object queries.

The framework introduces a spatial-semantic block that combines global semantic embedding with local spatial dependency modeling, which enhances the representation of target objects by capturing both broad context and fine details. Additionally, a masked cross-attention module refines object queries, concentrating on the most distinctive parts of target objects and reducing noise accumulation over time. This approach aids in effective long-term query propagation, a critical factor for high-performance tracking over extended sequences.

The experimental results are strong, showing state-of-the-art performance across several benchmarks.

**Strengths:**

This paper’s S3 algorithm for Video Object Segmentation (VOS) demonstrates notable strengths:

1.Spatial-Semantic Integration: By combining semantic embedding with spatial dependency modeling, it effectively captures complex object structures without requiring extensive ViT retraining.

2.Discriminative Query Mechanism: The adaptive query approach improves target focus and reduces noise in long-term tracking, enhancing robustness.

3.Extensive Validation: State-of-the-art results on multiple benchmarks highlight its strong generalization across datasets.

**Weaknesses:**

1.This paper claims to address the challenges of long-term tracking and segmentation. However, as far as I know, memory mechanisms are crucial for tackling these challenges in long-term tracking and segmentation, yet the authors do not seem to have conducted ablation experiments on the number of frames in the memory bank.

2.I believe that the ablation study on the number of queries is insufficient with only 8, 16, and 32 as tested values. A wider range of query counts should be explored to more thoroughly validate the effectiveness of the proposed method.

**Questions:**

See weakinesses

---

> ### Author Response · Authors · 2024-11-19
> **Response to Reviewer dBZh**
>
> We sincerely thank Reviewer dBZh for reviewing this paper.
>
> **Q1: Missing ablation study about the number of frames used in the long-term memory bank.**
> &nbsp;&nbsp; We present the experimental results of using different numbers of memorized frames in the following table. The experiments are conducted on the LVOS Val dataset, which shows greater sensitivity to the number of stored frames in the long-term memory.
>
> &nbsp;&nbsp; The table shows that the performance increases progressively with an increasing number of stored frames and the improvement saturates beyond a certain point. Considering the used Memory and run time also increase significantly, we set the Memory number as 20 (corresponding to 100 frames, storing one image every 5 frames), which achieves a good balance between accuracy and efficiency.
>
> | Memory Numbers | J&F  | J | F | Memory  | Inference Time   |
> |----------------|-------|--------|--------|---------|--------|
> | 10             | 68.9  | 64.3   | 73.5   | 2901M   | 39min  |
> | 20             | 69.3  | 64.8   | 74.0   | 4169M   | 40min  |
> | 30             | 70.1  | 65.4   | 74.8   | 6151M   | 46min  |
> | 40             | 71.1  | 66.3   | 76.0   | 8145M   | 52min  |
> | 50             | 71.2  | 66.2   | 76.1   | 9203M   | 57min  |
>
> **Q2: A wider range of query counts should be explored to more thoroughly validate the effectiveness of the proposed method.**
> &nbsp;&nbsp; In the table below, we also provide query numbers with additional values of 4, 10, and 64. The results show that:
>
> &nbsp;&nbsp;  (1) Increasing the number of queries does not consistently improve the model's performance. Using larger numbers of queries may degrade the performance slightly, as using too many queries will introduce more noise during training and slow down the convergence speed.
>
> | Queries | D17 (J&F) | MOSE-val (J&F) | YT19 (J&F) | LVOS (J&F) | YT18 (J&F) |
> |---------|-------------|----------------|------------|------------|------------|
> | 64      | 85.0        | 65.5           | 86.7       | 65.2       | 86.6       |
> | 32      | 85.8        | 68.3           | 86.9       | 64.8       | 86.8       |
> | 16      | 86.6        | 67.7           | 87.0       | 66.4       | 86.9       |
> | 10      | 86.4        | 68.2           | 87.0       | 66.4       | 87.1       |
> | **8**   | **86.7**    | **68.5**       | **87.5**   | **66.5**   | **87.4**   |
> | 4       | 83.1        | 60.2           | 85.2       | 53.5       | 85.2       |
>
> &nbsp;&nbsp; (2) The model achieves the best performance when the number of queries is set to 8. This is primarily because the number of queries has a close relationship with the number of targets. The table below presents the target number statistics across different datasets, which are mostly concentrated between 1 and 5. Setting the query number to 8 effectively covers almost all sequences in the datasets, as verified by the experimental results.
>
> | Target numbers | YTVOS19 | MOSE | LVOSv2 |
> |-------------------|---------|------|--------|
> | 1                 | 168     | 188  | 91     |
> | 2                 | 171     | 58   | 31     |
> | 3                 | 132     | 15   | 8      |
> | 4                 | 26      | 22   | 4      |
> | 5                 | 7       | 11   | 0      |
> | 6                 | 3       | 8    | 2      |
> | 7                 | 0       | 4    | 1      |
> | >=8               | 0       | 5    | 2      |

---

> ### Author Response · Authors · 2024-11-23
> **Please let us know if we address all the issues**
>
> Dear Reviewer,
>
> Thank you for the comments on our paper. We have provided a response and a revised paper on Openreview. Since the discussion phase ends on Nov 26, we would like to know whether we have addressed all the issues, and we look forward to resolving any additional questions or concerns you may have.
>
> Thank you again for your time and effort.
>
> Best regards

---

> ### Comment · Reviewer_dBZh · 2024-11-23
>
> Thank you for your response, which has basically resolved all my questions.

---

> > ### Author Response · Authors · 2024-11-25
> > **Thank you**
> >
> > Dear Reviewer,
> >
> > Since all the questions have been answered, could you consider raising the rating?
> >
> > Thank you,

---

### Official Review · Reviewer_TGSj · 2024-11-04

**Soundness:** 2
**Presentation:** 3
**Contribution:** 2
**Rating:** 6
**Confidence:** 4

**Summary:**

This paper focuses on video object segmentation. The authors analyze the existing challenges like structural complexity, occlusion, and dramatic appearance changes, and correspondingly propose spatial-semantic feature augmentation as well as discriminative query association. The ablation studies and visualizations verify the effectiveness of each module.

**Strengths:**

1. The motivation is clear and the architecture makes sense. Integrating high-level semantics and low-level spatial cues is promising in video object segmentation.
2. The experiments are thorough and the ablation studies can well reflect the effectiveness of each module.

**Weaknesses:**

1. The method is complicated. What is the advantage of using spatial offsets with deformable convolution compared to simple position encodings?
2. The second row of Figure 3(a) seems strange. With semantic feature augmentation, the feature maps can well highlight the desired object instance. Adding spatial cues on the contrary suppresses the emphasis on the target instance but enhances object instances with the same semantics.
3. Compared to SAM2, which designs a memory to prompt the segmentation of new frames, what is the advantage of this architecture?

**Questions:**

See weakness

---

> ### Author Response · Authors · 2024-11-19
> **Response to Reviewer TGSj**
>
> We sincerely thank Reviewer TGSj for reviewing this paper.
>
> **Q1: Deformable convolution or simple position encoding.**
> &nbsp;&nbsp; Compared to a simple position encoding mechanism, the deformable cross-attention module provides fine-grained spatial details and local features (using dynamic offsets), which better supports the association with the semantic features (especially for objects with complex structures). Besides, sparse attention with linear complexity is more suitable for VOS than global attention with quadratic complexity.
>
> &nbsp;&nbsp; In the table below, we replace deformable cross-attention with traditional cross-attention. The results demonstrate that using deformable cross-attention significantly enhances the model's performance, especially in datasets with complex targets (**\+1.2% gains over the MOSE dataset**).
>
> | Datasets              |  | D17|  | | MOSE |  |  | | YT19 |  |  |  | LVOS |  |
> |----------------------|-------------|-----------|-----------|----------------|--------------|--------------|------------|-----------|-----------|-----------|-----------|------------|----------|----------|
> | Methods               | J&F | J| F |J&F| J| F | J&F | Jseen| Fseen| Junseen | Funseen | J&F| J | F |
> | Global attention     | 86.1        | 82.0      | 90.2      | 67.3           | 63.2         | 71.5         | 86.4       | 85.5      | 90.2      | 81.1      | 88.7      | 64.1       | 59.5     | 68.7     |
> | Deformable attention | 86.7        | 82.7      | 90.8      | 68.5           | 64.5         | 72.6         | 87.5       | 86.8      | 91.8      | 81.3      | 89.9      | 66.5       | 62.1     | 70.8     |
>
> **Q2: Why does adding spatial cues suppress emphasis on the target instance while enhancing object instances with the same semantics?**
> &nbsp;&nbsp; During the spatial dependence modeling process, deformable attention is employed to allow the model to focus more on detailed information, which inevitably enhances the features of objects of the same class. This will not affect the target association across frames, since the proposed method develops a discriminative query propagation module to distinguish targets in the target association phase. In summary, the spatial cues contribute to more accurate predictions and the discriminative query propagation module ensures correct target associations.
>
> **Q3： Advantage compared to sam2.**
> &nbsp;&nbsp; Compared to SAM2, our approach has the following new designs.
>
> &nbsp;&nbsp; We introduce spatial-semantic information into VOS by explicitly modeling semantic and spatial information through the design of the SS Block, significantly enhancing the model's understanding of the target. Our model achieves comparable performance against SAM2 on various benchmarks, even when trained on a very small-scale dataset.
>
> &nbsp;&nbsp; To handle target association in long-term tracking scenarios, we propose a discriminative query mechanism, which contributes to the comparable performance against SAM2 (**\+7.9% in LVOSV2 and \+1.1% in LVOS V1**). The discriminative query generation effectively extracts and memorizes the salient feature points of the target, facilitating accurate identification and association during long-term tracking. The results on the LVOS dataset further validate this capability (**83.7% on LSVOSv2-val and 73.0% on LVOS-test**).

---

> ### Author Response · Authors · 2024-11-23
> **Please let us know if we address all the issues**
>
> Dear Reviewer,
>
> Thank you for the comments on our paper. We have provided a response and a revised paper on Openreview. Since the discussion phase ends on Nov 26, we would like to know whether we have addressed all the issues, and we look forward to resolving any additional questions or concerns you may have.
>
> Thank you again for your time and effort.
>
> Best regards

---

> > ### Comment · Reviewer_TGSj · 2024-11-25
> >
> > The rebuttal has addressed most of my concerns and I lean to accept.

---

> > > ### Author Response · Authors · 2024-11-26
> > > **Thanks**
> > >
> > > Dear Reviewer,
> > >
> > > We sincerely appreciate your acknowledgment and dedication in reviewing our work.
> > >
> > > Thank you

---

### Author Response · Authors · 2024-11-19
**General Response**

&nbsp;&nbsp; We thank the reviewers for their feedback and valuable suggestions, which helped us further strengthen the paper.

&nbsp;&nbsp; We are glad that all the reviewers found our approach to be **well-motivated** and **effective**, the integration of high-level semantics and low-level spatial cues to be **promising for VOS**, the **thoroughness of our experiments and ablation studies**, and the **state-of-the-art results** achieved on multiple VOS benchmarks. According to the comments, we have presented more details about the algorithm design and provided more experiments. We highlight a couple of common concerns in this general response and please find the detailed feedback under each comment.

### **(1) Experiments with a wider range of query counts to more thoroughly validate the effectiveness of the proposed method.**
&nbsp;&nbsp; In the table below, we provide results for query numbers 4, 10, and 64. The results show that increasing query numbers does not consistently improve performance, as more queries may introduce noise and cause slow convergence. The best performance is achieved with 8 queries, which are related to the target object number (mostly 1–5) across datasets, effectively covering most samples.
| Queries | D17 (J&F) | MOSE-val (J&F) | YT19 (J&F) | LVOS (J&F) | YT18 (J&F) |
|---------|-------------|----------------|------------|------------|------------|
| 64      | 85.0        | 65.5           | 86.7       | 65.2       | 86.6       |
| 32      | 85.8        | 68.3           | 86.9       | 64.8       | 86.8       |
| 16      | 86.6        | 67.7           | 87.0       | 66.4       | 86.9       |
| 10      | 86.4        | 68.2           | 87.0       | 66.4       | 87.1       |
| **8**   | **86.7**    | **68.5**       | **87.5**   | **66.5**   | **87.4**   |
| 4       | 83.1        | 60.2           | 85.2       | 53.5       | 85.2       |

| Target numbers | YTVOS19 | MOSE | LVOSv2 |
|-------------------|---------|------|--------|
| 1                 | 168     | 188  | 91     |
| 2                 | 171     | 58   | 31     |
| 3                 | 132     | 15   | 8      |
| 4                 | 26      | 22   | 4      |
| 5                 | 7       | 11   | 0      |
| 6                 | 3       | 8    | 2      |
| 7                 | 0       | 4    | 1      |
| >=8               | 0       | 5    | 2      |

### **(2) More ablation experiments about the Spatial-Semantic block.**
&nbsp;&nbsp; To save space, we placed the detailed ablation study in Table 7 in the Appendix section. Table 7 shows that spatial dependence modeling significantly improves performance across all VOS datasets, with a 3.0%+ J&F gain on MOSE val. This improvement comes from better detail modeling during feature extraction. Adding semantic information further boosts performance, reaching 73% J&F on the LVOS test, demonstrating its value in enhancing target semantic understanding. We provide some experiments in the table below. For more details, please refer to Table 7 in the  Appendix.

| **Dataset**                        | **MOSE** | **D17 test** | **LVOS test** | **YT19** |
|------------------------------------|----------|--------------|---------------|----------|
| **Method**                         | **J&F**  | **J&F**      | **J&F**       | **J&F**  |
| **Trained on YouTubeVOS and DAVIS datasets** |          |              |               |          |
| XMem (Baseline)                    | 53.3     | 81.0         | 50.0          | 85.5     |
| Cutie                              | 64.0     | 84.2         | 56.2          | 86.1     |
| +Discriminative Query               | 64.2     | 85.2         | 57.4          | 86.5     |
| +Spatial                            | 68.2     | 86.2         | 67.4          | 87.3     |
| +Semantic (Full)                    | 68.5     | 86.7         | 66.5          | 87.5     |
| **Trained on the MEGA datasets**   |          |              |               |          |
| Cutie                              | 69.9     | 86.1         | 66.7          | 87.0     |
| +Discriminative Query               | 70.6     | 86.6         | 66.5          | 87.5     |
| +Spatial                            | 73.5     | 87.6         | 68.8          | 87.9     |
| +Semantic (Full)                    | 74.0     | 87.8         | 73.0          | 88.1     |

---

### Meta-Review · Area_Chair_TE13 · 2024-12-21

**Metareview:**

This paper proposes a robust video object segmentation framework by learning spatial-semantic features and discriminative object queries to address challenges encountered in the video object segmentation task. The overall idea seems like a combination of existing techniques, yet experimental results are solid. Four reviewers reach a consensus to accept this paper by discussing it with the authors through rebuttals. After reviewing the comments and rebuttals, AC agrees with the merits proposed in this paper. Therefore, AC recommends accepting this paper.

**Additional Comments On Reviewer Discussion:**

All main concerns (e.g., deformable convolution used, advantages with SAM2, missed ablations on the number of frames, comparisons with jointformer, etc.) are addressed by the rebuttal. AC also agrees with the responses of the authors in addressing the concerns. Also, please update the responses in the final version.

---

### Decision · Program_Chairs · 2025-01-22

Accept (Poster)